# Lung gene expression signatures suggest pathogenic links and molecular markers for pulmonary tuberculosis, adenocarcinoma and sarcoidosis

Qiyao Chai[1,2,5], Zhe Lu[1,2,5], Zhidong Liu[3,5], Yanzhao Zhong[1,2], Fuzhen Zhang[1], Changgen Qiu[1,2], Bingxi Li[1], Jing Wang[1], Lingqiang Zhang [4], Yu Pang[3✉] & Cui Hua Liu[1,2✉]

Previous reports have suggested a link between pulmonary tuberculosis (TB), which is caused by *Mycobacterium tuberculosis* (Mtb), and the development of lung adenocarcinoma (LUAD) and sarcoidosis. Furthermore, these lung diseases share certain clinical similarities that can challenge differential diagnosis in some cases. Here, through comparison of lung transcriptome-derived molecular signatures of TB, LUAD and sarcoidosis patients, we identify certain shared disease-related expression patterns. We also demonstrate that *MKI67*, an over-expressed gene shared by TB and LUAD, is a key mediator in Mtb-promoted tumor cell proliferation, migration, and invasion. Moreover, we reveal a distinct ossification-related TB lung signature, which may be associated with the activation of the BMP/SMAD/RUNX2 pathway in Mtb-infected macrophages that can restrain mycobacterial survival and promote osteogenic differentiation of mesenchymal stem cells. Taken together, these findings provide novel pathogenic links and potential molecular markers for better understanding and differential diagnosis of pulmonary TB, LUAD and sarcoidosis.

[1] CAS Key Laboratory of Pathogenic Microbiology and Immunology, Institute of Microbiology, Center for Biosafety Mega-Science, Chinese Academy of Sciences, 100101 Beijing, China. [2] Savaid Medical School, University of Chinese Academy of Sciences, 101408 Beijing, China. [3] Beijing Tuberculosis and Thoracic Tumor Research Institute, Beijing Chest Hospital, Capital Medical University, 101149 Beijing, China. [4] State Key Laboratory of Proteomics, Beijing Proteome Research Center, National Center of Protein Sciences (Beijing), Beijing Institute of Lifeomics, 100850 Beijing, China. [5]These authors contributed equally: Qiyao Chai, Zhe Lu, Zhidong Liu. ✉email: pangyupound@163.com; liucuihua@im.ac.cn

Tuberculosis (TB), a communicable disease caused by *Mycobacterium tuberculosis* (Mtb), remains a major cause of morbidity and mortality worldwide[1]. Mtb is a paradigmatic intracellular pathogen that can persist in the lung and interact with host cells for a long term, leading to a range of pathological outcomes[2,3]. TB infection has long been associated with multiple other lung diseases, including lung cancer and sarcoidosis[4–6]. Internationally, lung cancer is the leading cause of cancer-related death in humans, among which lung adenocarcinoma (LUAD) is the most common histologic subtype[7]. Pre-existing TB significantly increases the incidence of lung cancer, particularly LUAD[8,9]. In turn, lung cancer is also a strong risk factor for TB[10], implying interactive effects between TB infection and lung cancer. Sarcoidosis is regarded as a multisystem autoimmune disease that preferentially affects the lungs and lymph glands and has recently been linked to TB infection. Both diseases could form similarly well-organized granulomatous lesions, and mycobacterial genetic material and antigens can be found in a considerable proportion of sarcoidosis patients, suggesting that infection with mycobacteria, such as Mtb, might be an important pathophysiologic factor for sarcoidosis[6,11]. More importantly, pulmonary TB could mimic lung cancer[12–15] or sarcoidosis[6,16] in some cases, which may challenge the diagnosis and delay the treatment. Therefore, unfolding the pathogenetic links and differences among TB, LUAD and sarcoidosis can help better understand Mtb pathogenesis and clinical outcomes, as well as benefit differential diagnosis of these clinically important lung diseases.

Previous studies on blood transcriptional profiling of TB patients have provided enlightening insights into TB immunopathogenesis and diagnosis[17–19]. However, blood-based immunodiagnostic tests are inevitably influenced by the change of global host immune status[20]. Moreover, TB blood modular signature shows close resemblance with that of some other infectious and inflammatory lung diseases[21–23], perhaps because the blood transcriptome could not adequately represent disease-specific pathogenic changes in the lung microenvironment. Therefore, in this study, we performed transcriptional profiling of lung tissues from TB, LUAD, and sarcoidosis patients with an aim to better reveal local pathogenic characteristics of these diseases. On the one side, we reveal that TB shares a set of potential pathogenic mediators with LUAD or sarcoidosis, and further demonstrate that host *MKI67*, an over-expressed gene shared by TB and LUAD, is involved in Mtb-promoted tumor cell proliferation, migration, and invasion. On the other side, we unravel distinct lung modular signatures and molecular markers of TB, LUAD, and sarcoidosis, and verify an ossification-related TB lung signature, which is probably associated with the activation of BMP/SMAD/RUNX2 pathway in Mtb-infected macrophages that controls mycobacterial survival and promotes osteogenic differentiation of mesenchymal stem cells (MSCs). Our findings provide new insights into Mtb pathogenesis and may contribute to differential diagnosis of TB, LUAD, and sarcoidosis.

## Results

**Lung transcriptome-derived molecular signatures unravel pathogenic links among TB, LUAD, and sarcoidosis.** A total of 104 patients including 35 TB, 48 LUAD, and 21 sarcoidosis patients from mainland China who received lung resection surgery or diagnostic biopsy were recruited for collection of lung samples and the corresponding patient demographic and clinical characteristics (Supplementary Data 1). To investigate the pathogenic links among TB, LUAD and sarcoidosis, we started by generating lung molecular signatures based on genome-wide transcriptional profiling of TB, LUAD, sarcoidosis and normal

control (NC) lung tissues ($n = 5$ of each group) using RNA-Seq technology (Supplementary Table 1). Differentially expressed genes (DEGs) of each disease group were then derived by comparing to NC (Supplementary Data 2–4). According to Venn diagram analysis, a total of 192 overlapping DEGs among TB, LUAD and sarcoidosis groups were identified (Fig. 1a), which represent common changes in lung transcriptional profiles among these diseases. Hierarchical clustering followed by Reactome pathway enrichment analysis of these transcripts indicated that TB, LUAD, and sarcoidosis were largely correlated with over-represented extracellular matrix (ECM) organization-associated pathways such as those involved in collagen degradation, assembly of collagen fibrils and other multimeric structures, as well as collagen biosynthesis and modifying enzymes (hypergeometric test, $P < 0.001$) (Fig. 1b). We next investigated the similar transcriptional changes between TB and LUAD groups, through combining lung transcriptional data of LUAD and control samples ($n = 510$ and 58, respectively) from The Cancer Genome Atlas (TCGA) (https://portal.gdc.cancer.gov/). Using Kaplan–Meier analysis, we identified a total of 65 TB-LUAD-shared DEGs (including 20 upregulated and 45 downregulated genes) whose expression was significantly correlated with the overall survival (OS) of LUAD patients (log-rank $P < 0.05$) from TCGA (Fig. 1c and Supplementary Data 5). Gene functional enrichment analysis indicated that these genes were dominantly associated with ECM organization, immune response, cell growth and proliferation and metabolic process (Fig. 1c, d), which could be the underlying molecular pathologic links between TB infection and LUAD development, and these genes were potentially the key LUAD-correlated pathogenic mediators that affected by TB infection. Likewise, we further identified a 20-gene similar lung signature between TB and sarcoidosis patients using combined analysis of our RNA-Seq data and the published transcriptional data of sarcoidosis lung samples with the matched controls (both $n = 6$) from a microarray study[24] (Fig. 1e and Supplementary Data 6). Functional interpretations indicated the association of these 20 transcripts with ECM organization, immune response, and cell migration (Fig. 1e, f). Together, these results reveal certain similar transcriptional changes involving shared biological processes among TB, LUAD, and sarcoidosis patients, indicating potential pathogenic links among these diseases.

**TB patients share potential pathogenic mediators with LUAD and sarcoidosis patients.** To verify the transcriptional similarity among TB, LUAD, and sarcoidosis patients that showed over-represented biological pathways related to collagen matrix remodeling, we performed histological analysis of lung sections from each sample group. Masson's trichrome staining suggested that all these diseases drove pulmonary remodeling with various degrees of matrix destruction and fibrous protein deposition (Fig. 2a). Immunohistochemical analysis further confirmed that collagen I and collagen III, the representative lung interstitial ECM collagens[25], were similarly over-abundant in TB, LUAD, and sarcoidosis lungs as compared with the controls (Fig. 2a–c). Therefore, the collagen matrix remodeling in the lung is perhaps a common pathogenic characteristic of TB, LUAD, and sarcoidosis. We then sought to confirm potential pathogenic mediator genes of LUAD or sarcoidosis whose expression were also affected by TB infection. Among 65 similar signature genes between TB and LUAD (see Fig. 1c, d), those genes involved in cell growth and proliferation may be most directly correlated with tumor development. We thus focused on *MKI67*, which encodes Ki-67, a nuclear DNA-binding protein exclusively expressed in proliferating cells and thus widely used as a proliferation marker in

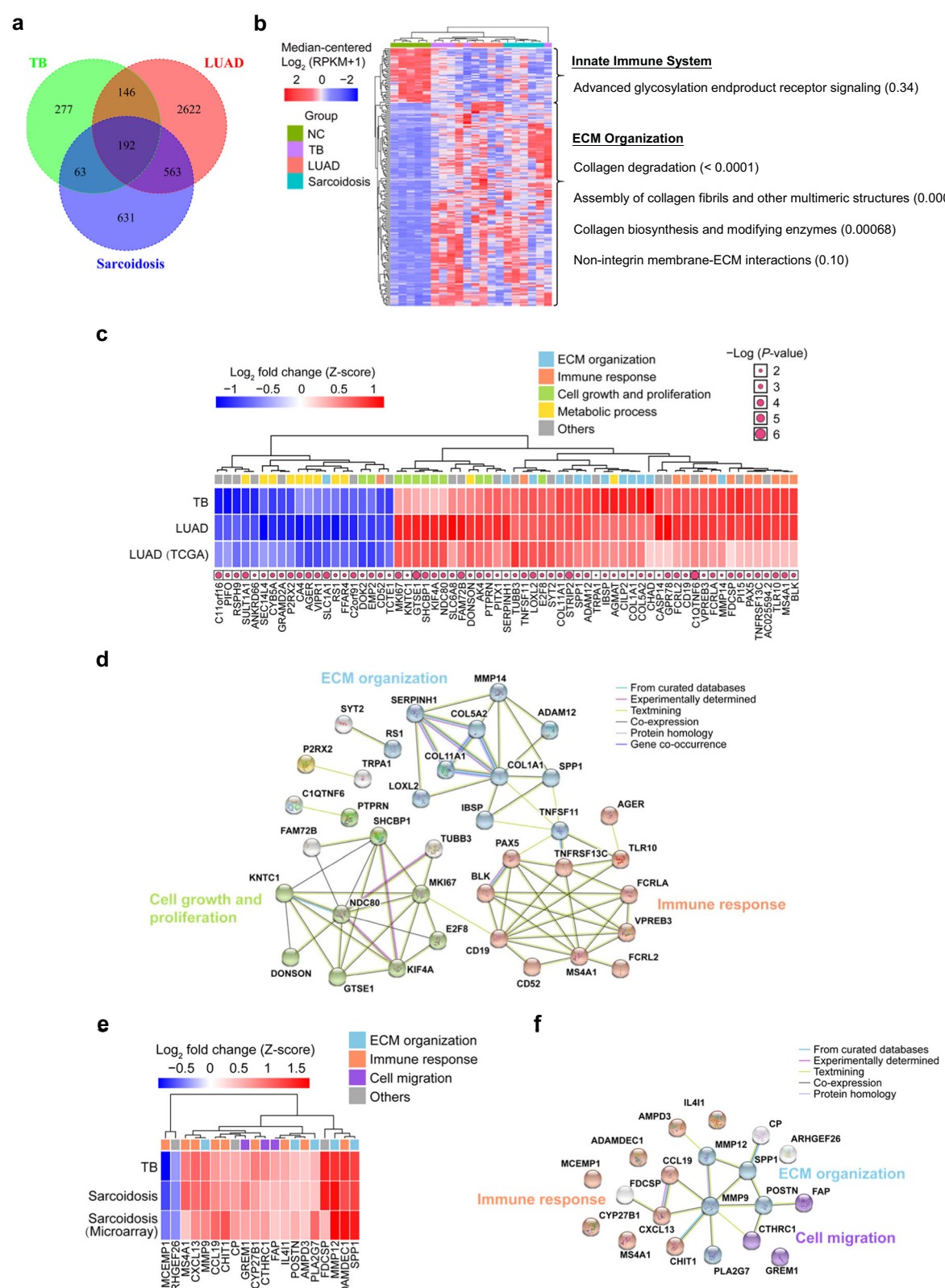

cancer prognosis[26], and we consistently found that the over-expression of *MKI67* was correlated with reduced OS of LUAD patients according to Kaplan–Meier analysis based on TCGA data (log-rank $P = 0.0002$) (Fig. 2d). Through quantitative PCR (qPCR) analysis of lung samples (Supplementary Table 2 and Supplementary Data 7), we found that *MKI67* mRNA were similarly increased in TB and LUAD groups as compared to NC

group (Fig. 2e), consistent with the results from gene expression profiles. Immunohistochemical analysis further confirmed over-expression of Ki-67 in both TB and LUAD lungs (Fig. 2f, g). These results indicate a common feature of enhanced cell proliferation in lung lesions of TB and LUAD. Among 20 similar signature genes between TB and sarcoidosis (see Fig. 1e, f), we noticed that *MMP12* and *ADAMDEC1*, two previously reported

**Fig. 1 Lung molecular signatures reveal pathogenic links among TB, LUAD, and sarcoidosis. a** Venn diagram depicting the number and the overlap of differentially expressed genes of TB, LUAD, and sarcoidosis lung tissues as compared to normal control (NC) lung tissues. **b** Heatmap showing two dominant clusters of the 192 overlapped differentially expressed genes of TB, LUAD, and sarcoidosis lung tissues by Reactome pathway enrichment analysis. The enriched gene pathways are listed on the right, with Benjamini- Hochberg (BH)-corrected *P*-values in parentheses (hypergeometric test). **c** Heatmap depicting 65 lung-tissue transcripts that were similarly expressed in TB, LUAD, and LUAD from TCGA groups. All genes shown were significantly correlated with the overall survival (OS) of lung LUAD patient (log-rank test *P* < 0.05, as indicated by pink dots), according to Kaplan–Meier analysis based on TCGA data. Functional interpretations of genes are indicated by color-coded squares according to STRING functional enrichment analysis. **d** Interaction network analysis of genes as in **c**. Genes with functions related to extracellular matrix (ECM) organization, immune response, cell growth and proliferation and metabolic process as annotated in **c** are colored in blue, red, green, and yellow, respectively. **e** Heatmap depicting 20 lung-tissue transcripts that were similarly expressed in TB, sarcoidosis, and sarcoidosis from microarray data groups. Functional interpretations of genes are indicated by color-coded squares according to STRING functional enrichment analysis. **f** Interaction network analysis of genes as in **e**. Genes involved in the biological pathways associated with immune response, ECM organization, and cell migration as annotated in **e** are colored in red, blue, and purple, respectively.

potential pathogenic mediator genes of sarcoidosis[24], were over-represented in both TB and sarcoidosis samples, which result was further confirmed by qPCR analysis (Fig. 2h, i). Likewise, we also verified the over-expression of several other inflammation-associated genes including *CCL19*, *CXCL13*, and *CYP27B1* in the lung of both TB and sarcoidosis patients (Fig. 2j–l). In summary, identification of these potential pathogenic mediators of LUAD or sarcoidosis shared by TB provides clues for probable molecular mechanisms underlying the pathogenic links between Mtb infection and LUAD or sarcoidosis.

***MKI67* is an important mediator for Mtb-induced tumor cell proliferation, migration, and invasion**. Next, we wondered how Mtb manipulates those potential pathogenic mediators and facilitates the corresponding disease outcomes. Interestingly, according to our previous findings, *MKI67* is a potential host gene regulated by Mtb PtpA, a key effector protein that can enter into the host cell nucleus and regulate cell proliferation and migration to promote tumor development[27]. Since *MKI67* was over-expressed in both TB and LUAD lungs, we speculated that host *MKI67* could be an important mediator of tumor cell proliferation and migration that is employed by Mtb during infection. To test this hypothesis, we started by infecting human lung adenoma A549 cells with Mtb H37Rv and found that *MKI67* mRNA was increased during the course of Mtb infection (Supplementary Fig. 1a). We then conducted Cell Counting Kit-8 (CCK-8) assay to show that Mtb could promote the proliferation of the control A549 cells at 24 and 48 h post-infection, which effect was largely abolished by Ki-67 knock-down (Supplementary Fig. 1b, c). We further confirmed this result through cell proliferation assay using 5-ethynyl-2′-deoxyuridine (EdU), a nucleoside analogue, which can be incorporated into the DNA of proliferating cells[28] (Fig. 3a, b). Moreover, transwell culture assays demonstrated that Mtb infection significantly increased the migration and invasion of A549 cells, which effects were markedly attenuated by Ki-67 knock-down (Fig. 3c–f). Notably, mycobacteria were carried by infected A549 cells to move through the transwell membrane as well (Fig. 3c, e). Thus, Mtb promotes the proliferation, migration, and invasion of A549 cells partially depending on *MKI67*.

Since *MKI67* is a potential host target of Mtb effector PtpA, we then further examined the role of PtpA in *MKI67*-mediated cell proliferation, migration, and invasion during Mtb infection. And we found that deletion of PtpA in Mtb largely attenuated the effect of Mtb infection-induced upregulation of *MKI67* mRNA (Fig. 3g) and Ki-67 protein in A549 cells (Fig. 3h, i). Consistently, PtpA deletion also markedly abolished the effects of Mtb infection-promoted A549 cell proliferation, migration, and invasion, which were largely dependent on Ki-67 (Supplementary Fig. 1d–j). Together, these results indicate that Mtb effector protein PtpA contributes to Mtb-promoted tumor cell

proliferation, migration, and invasion, which are partially dependent on *MKI67*.

**TB, LUAD, and sarcoidosis are characterized by distinct modular signatures**. To further reveal the differences in molecular pathogenesis among TB, LUAD, and sarcoidosis to identify potential biomarkers, we then analyzed specific lung molecular signatures of these diseases using a modular approach based on weighted gene co-expression network analysis (WGCNA)[29]. WGCNA divided genes (*n* = 16,298) from the transcriptional profiles of NC, TB, LUAD, and sarcoidosis samples into 27 modules (Fig. 4a and Supplementary Fig. 2). This approach uncovered that modules 4, 21, and 26 were the most specific gene modules correlated with TB (correlation test, *P* = 0.0074), LUAD (*P* = 0.0002), and sarcoidosis (*P* = 0.0395), respectively (Supplementary Fig. 2). We then performed gene ontology (GO) enrichment analysis for each gene module (Fig. 4a and Supplementary Data 8). The modular signature of TB (module 4) showed over-represented genes related to ECM organization or ossification, as compared to the other two diseases (Fig. 4a, b). Likewise, the modular signature of LUAD (module 21) and sarcoidosis (module 26) showed over-represented genes involved in cell proliferation and oxidation-reduction process, respectively, as compared to the others (Fig. 4a, c, d). In summary, these results present an overview of the differences in lung molecular characteristics among TB, LUAD, and sarcoidosis, and provide distinct modular signatures of each disease.

**TB, LUAD, and sarcoidosis can be differentiated by specific pathogenic markers in the lung**. Next, we sought to identify certain disease-specific molecular markers using the modular signatures. TB-specific signature (module 4) showed over-represented genes involved in ECM component organization and interaction, or ossification-related processes (Supplementary Fig. 3a), both of which are closely associated with lung ECM remodeling. Gene set enrichment analysis (GSEA) consistently revealed a positive correlation of TB patients with ECM organization-related transcriptional signature (Supplementary Fig. 3b). ECM proteases, such as matrix metalloproteinases (MMPs), play a central role in the regulation of lung matrix destruction and reorganization during TB infection[2,30]. Given that TB exhibited a distinct ECM remodeling signature as compared to LUAD and sarcoidosis, we wondered if there is any specific ECM proteases differentially expressed in TB lungs to drive this process. The results from TB lung transcriptional profiling (Supplementary Data 2), combined with previous studies[30], suggested that multiple members of MMPs, including MMP1, MMP2, MMP3, MMP7, MMP8, MMP9, MMP10, MMP11, MMP12, MMP13, and MMP14, were potentially involved in TB-induced lung remodeling. In addition, we also

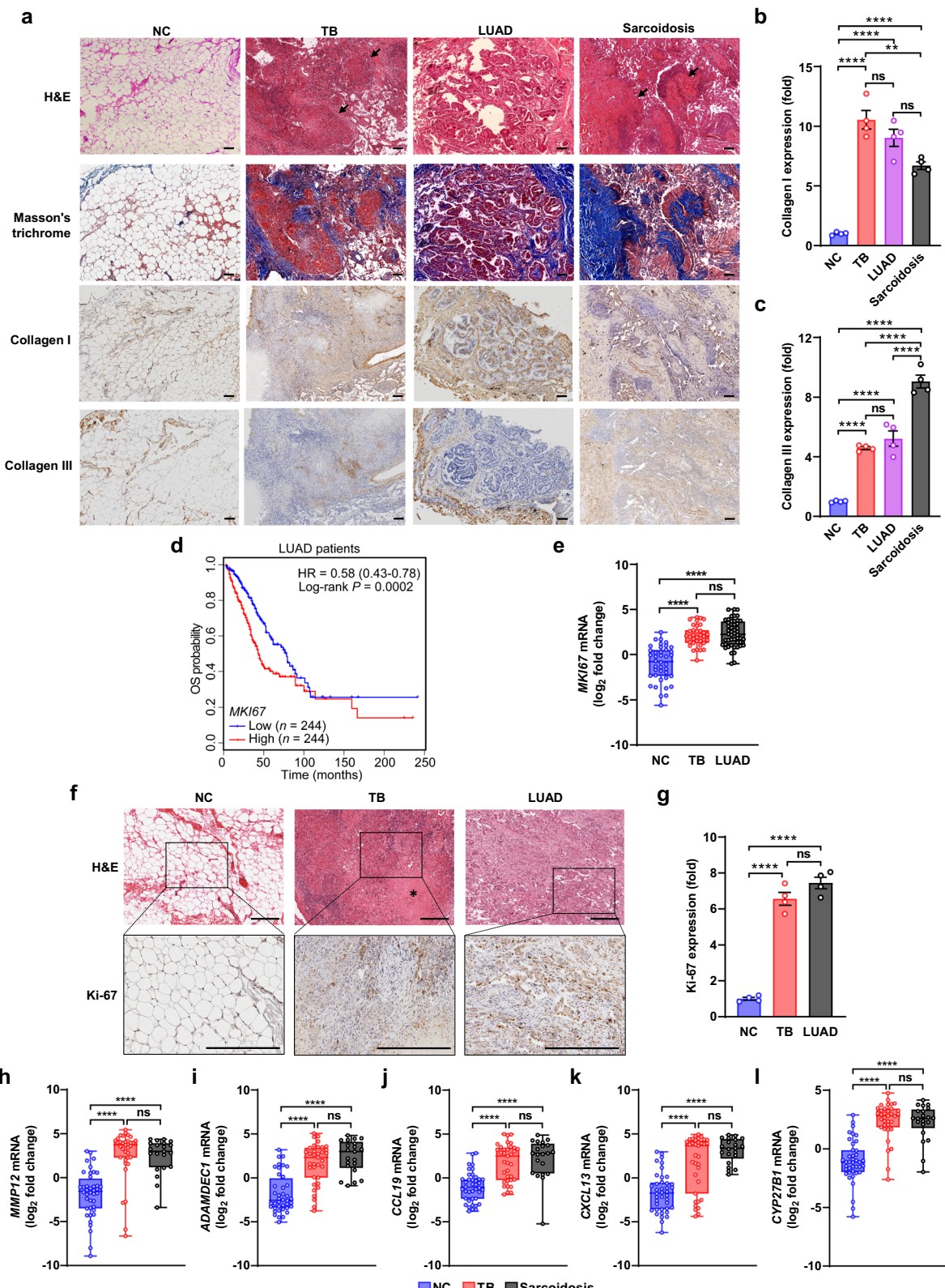

noticed that a considerable number of TB DEGs encode other ECM protease family members, such as *CTSK*, *ADAMs*, *ADAMTSs*, and *PRSSs* (Supplementary Data 2). Therefore, we performed qPCR analysis of these genes and found that only *CTSK* mRNA and *MMP8* mRNA were specifically increased in TB sample group (Fig. 5a, b and Supplementary Fig. 4). Immunohistochemical analysis further confirmed over-expression of

these two gene products, Cathepsin K and MMP8, in TB lungs, but only MMP8 was specifically increased in TB lungs as compared to the other groups (Fig. 5c–e). Interestingly, MMP8 was dominantly expressed in the necrotizing foci of TB (Fig. 5c). These results indicate that MMP8 is potentially a TB-specific ECM protease, and is closely related to TB-induced lung tissue degradation.

**Fig. 2 TB patients share certain pathogenic mediators with LUAD and sarcoidosis patients. a** Representative images of histologic and immunohistochemical analysis of collagen fibers in NC, TB, LUAD, and sarcoidosis lung sections ($n = 8$, 10, 12, and 8, respectively). Fibrous matrix deposits (blue) were identified by Masson's trichrome staining. Arrows indicate granulomatous lesions. Scale bars, 100 μm. **b, c** Quantitative analysis of collagen I (**b**) and collagen III (**c**) expression as in **a**. Four independent visual fields were examined (mean ± s.e.m. of $n = 4$). $P > 0.05$, not significant (ns); **$P < 0.01$; ****$P < 0.0001$ (one-way ANOVA). **d** The overall survival (OS) of lung LUAD patients from TCGA was compared between individuals with high or low levels of *MKI67* mRNA transcription. Hazard ratio (HR), 95% confidence interval and log-rank $P$-values are shown. **e** Quantitative PCR analysis of *MKI67* mRNA in NC, TB and LUAD groups. **f** Representative images of histologic and Ki-67 immunohistochemical analysis of NC, TB, and LUAD lung sections ($n = 8$, 10, and 12, respectively). The asterisk indicates the necrotizing focus. Scale bars, 200 μm. **g** Quantitative analysis of Ki-67 expression as in **f**. Four independent visual fields were examined (mean ± s.e.m. of $n = 4$). $P > 0.05$, not significant (ns); ****$P < 0.0001$ (one-way ANOVA). **h–l** Quantitative PCR analysis of *MMP12* (**h**), *ADAMDEC1* (**i**), *CCL19* (**j**), *CXCL13* (**k**), and *CYP27B1* (**l**) mRNAs in NC, TB, and sarcoidosis groups. For **e**, **h–l**, Box-whisker plot indicates the interquartile range (box), the median value (line within the box) and the maximum and minimum value (whiskers). $P > 0.05$, not significant (ns); ****$P < 0.0001$ ($n = 40$, 35, 48, and 21 in NC, TB, LUAD, and sarcoidosis groups, respectively, one-way ANOVA). Results are representatives from three independent experiments.

LUAD-specific signature (module 21) exhibited over-abundance of biological processes related to cell proliferation, such as cell division, DNA replication, etc. (Supplementary Fig. 5a). Pathway enrichment followed by interaction network analysis further showed a majority of LUAD signature genes were involved in cell cycle (Supplementary Fig. 5b, c). Notably, *BRCA1* and *PCNA* were at the center of the interaction network of LUAD signature genes, and these two genes were significantly correlated with the survival of LUAD patients according to Kaplan–Meier analysis (log-rank $P = 0.0012$ and 0.0030, respectively) (Supplementary Fig. 5c–e). We thus performed qPCR analysis to verify the expression of them and found that both *BRCA1* and *PCNA* mRNAs were specifically increased in LUAD lungs (Fig. 5f, g). Immunohistochemical analysis further confirmed the relatively specific over-expression of BRCA1 and PCNA in LUAD lungs as compared to the other groups (Fig. 5h–j). Therefore, BRCA1 and PCNA were potentially the specific lung molecular markers of LUAD.

Pulmonary sarcoidosis-specific signature (module 26) exhibited over-abundance of genes involved in oxidation-reduction process (Supplementary Fig. 6a). Pathway enrichment analysis of sarcoidosis transcriptional profile consistently showed the enrichment of various metabolic pathways (Supplementary Fig. 6b), amongst which, we focused on the arachidonic acid (AA) metabolism pathway, since it generates various metabolic intermediates that play crucial regulatory roles in local immune homeostasis and are associated with multiple inflammatory diseases[31]. Interaction network analysis further uncovered a considerable number of sarcoidosis signature genes encoding catalyzing enzymes or the receptors of intermediates in AA metabolism (Supplementary Fig. 6c). We thus performed qPCR-based transcriptional analysis of the key enzymes and receptors in AA metabolism[32,33], and found that a majority of these molecules were over-represented in sarcoidosis, but not in TB or LUAD, as compared to NC (Supplementary Figs. 7 and 8). According to Gini score calculated from gene expression values, we examined the top-6 most important AA metabolism-related genes (including *PLA2G6*, *PLA2G7*, *AKR1C1*, *AKR1C3*, *LTA4H*, and *PTGER4*) in sarcoidosis, whose expression showed a positive correlation with sarcoidosis samples (Fig. 5k). Hierarchical clustering analysis using this 6-gene signature showed that sarcoidosis samples could automatically cluster together (Fig. 5l). Further receiver operating characteristic (ROC) curve analysis suggested that these 6 genes probably had a good predictive ability (AUC ≥ 0.872; DeLong test, $P < 0.001$) to distinguish sarcoidosis samples from NC and the other disease samples (Fig. 5m). Taken together, certain lung pathogenic markers identified from distinct modular signatures of TB, LUAD, and sarcoidosis may benefit the differential diagnosis of these pulmonary diseases.

**Mtb activates BMP/SMAD/RUNX2 signaling pathway and induces pulmonary ossification in TB patients.** Consistent with the finding of TB lung signature relating to ossification, a large proportion of recruited TB patients (51.4%) were featured with computed tomography (CT) findings suggestive of calcium deposition in the lung (Supplementary Data 1 and Supplementary Table 2), reminiscent of a clinical link between TB infection and subsequent pulmonary calcification and ossification[34,35]. We thus attempted to investigate the mechanism underlying Mtb-induced lung ossification, which has long been overlooked. GSEA and hierarchical clustering analysis of TB and NC transcriptional profiles revealed numerous ossification-related genes over-expressed in TB lungs, such as *RUNX2*, a vital transcriptional regulator of osteoblast differentiation and bone formation, and osteogenic marker genes *IBSP*, *SPP1*, *POSTN*, *BGLAP*, and *SPARC* (Supplementary Fig. 9a, b). We then divided TB samples into noncalcified and calcified groups based on chest CT results and performed qPCR analysis to verify the expression of these genes in TB, and the data confirmed the increased expression of them in TB samples, particularly in calcified TB samples, as compared to NC (Supplementary Fig. 9c–h). These results indicate that TB lungs, particularly those with calcification, were closely associated with ossification processes. To further confirm this result, we selected two representative lung samples from calcified TB patients with (TB-7) or without (TB-3) cavitation for histological analysis (Fig. 6a and Supplementary Data 1). Patient TB-7 showed a massive aggregation of fibrous matrices with dispersed deposits of calcium around the necrotizing foci of TB granulomas (Fig. 6b). By comparison, patient TB-3 showed the formation of bone spicule and the mass deposits of calcium in the abnormal osteoid within lung lesions, where the typical bone cells including osteocytes, osteoblasts and osteoclast-like giant cells can also be found (Fig. 6b). Since *RUNX2* mRNA was increased in TB lungs, we further performed immunohistochemical analysis to examine the abundance of RUNX2 and p-SMAD1/5/9 for indicating the activation of bone morphogenetic protein (BMP)/SMAD signaling pathway, which regulates RUNX2 expression and plays a crucial role in osteogenesis[36,37]. As compared to NC, both RUNX2 and p-SMAD1/5/9 were significantly increased in the lung of the two calcified TB patients, particularly in non-cavitating patient TB-3 (Fig. 6c–e). These histological observations were consistent with the CT findings that the calcification in TB lesion of patient TB-3 was much more prominent than that in patient TB-7. We next examined additional lung samples and further confirmed an increased p-SMAD1/5/8 and RUNX2 in calcified TB lungs as compared to NC (Fig. 6f). Therefore, calcified TB lungs were closely associated with the activation of ossification-related signaling pathways.

To further determine whether Mtb infection could activate BMP/SMAD signaling pathway and increase the expression of

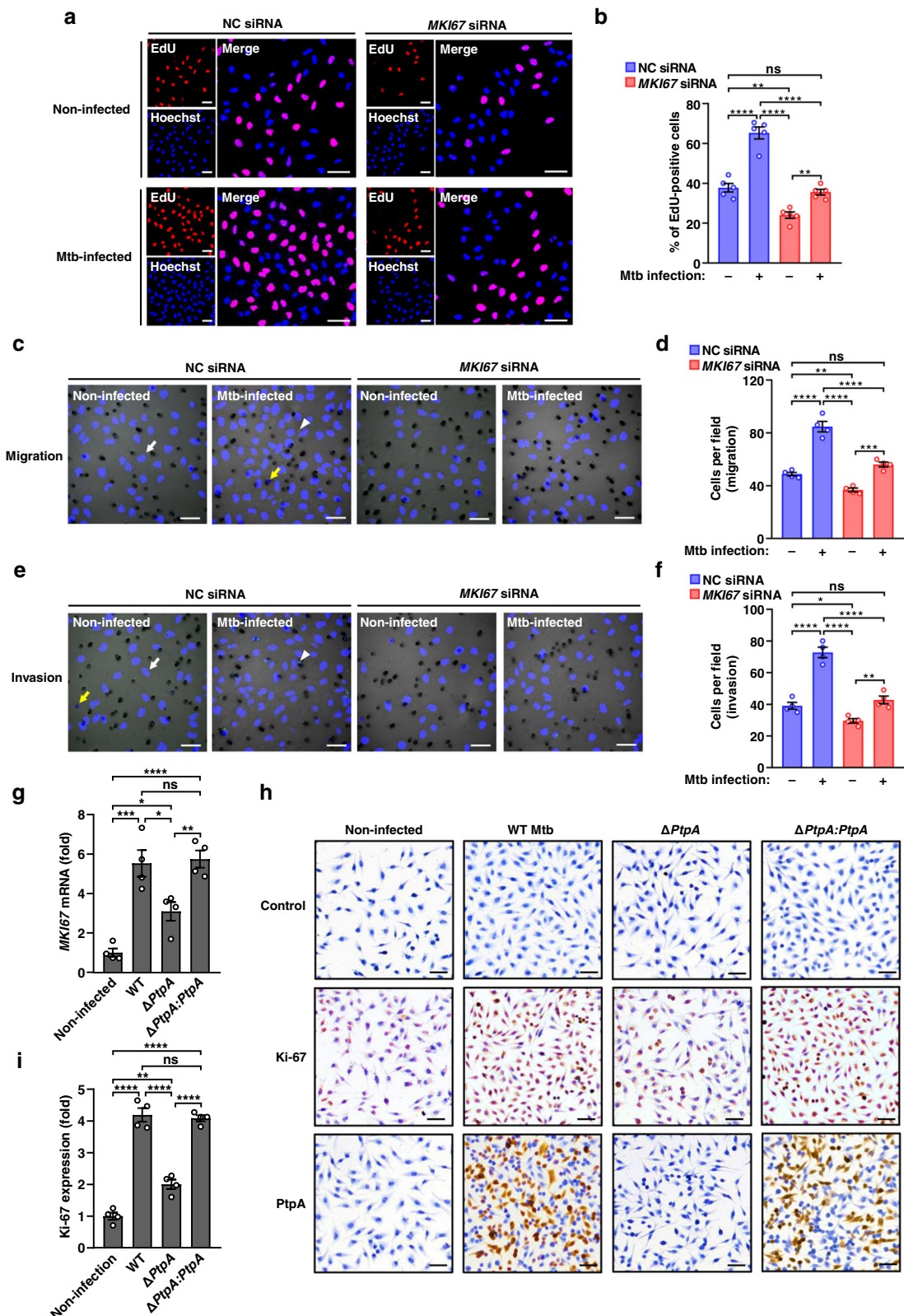

downstream RUNX2, human macrophage-like U937 cells were infected with Mtb. Expectedly, Mtb infection could increase BMP2/4, p-SMAD1/5/9, and RUNX2 in macrophages (Fig. 6g). Mtb-induced increase of BMP2/4, p-SMAD1/5/9, and RUNX2 were suppressed when U937 cells were treated with LDN-193189 or K02288 (which are specific inhibitors of BMP type I receptors[38,39]), but not Galunisertib (which is a TGFβ type I receptor inhibitor[40]) (Fig. 6g). Therefore, Mtb could induce activation of BMP/SMAD signaling, rather than TGF β-mediated non-SMAD signaling, to increase RUNX2 expression in host

**Fig. 3 Mtb promotes tumor cell proliferation, migration and invasion partially depending on MKI67. a** Representative images of EdU proliferation assays of A549 cells. Cells were transfected with NC or *MKI67* siRNA for 24 h, and infected with or without Mtb for 24 h. The proliferating cells were labeled with incorporated EdU-594 (red), and nuclei were stained with Hoechst 33342 (blue). Scale bars, 50 μm. **b** Quantification of EdU-positive cells as in **a**. Five independent visual fields were examined. **c, e** Transwell migration (**c**) and invasion (**e**) assays of A549 cells. Cells were transfected with NC or *MKI67* siRNA for 24 h and infected with or without Mtb, and were then allowed to migrate or invade for 12 h. The white arrow indicates the cell (stained with DAPI, blue) that had moved through the filter, and the yellow arrow indicates the cell that was still moving through a pore. The arrowhead indicates Mtb (stained with Alexa Fluor 488 succinimidyl ester, green). Scale bars, 50 μm. **d, f** Quantification of cells that had migrated (**d**) or invaded (**f**) through the filter. Four independent visual fields were examined. **g** Quantitative PCR analysis of *MKI67* mRNA in A549 cells. Cells were infected with the indicated Mtb strains or not for 48 h. **h** Representative images of Ki-67 and PtpA immunohistochemical analysis of A549 cells treated as in **g**. Scale bars, 50 μm. **i** Quantitative analysis of Ki-67 expression as in **h**. Four independent visual fields were examined. $P > 0.05$, not significant (ns); $*P < 0.05$; $**P < 0.01$; $***P < 0.001$; $****P < 0.0001$ (mean ± s.e.m. of $n = 5$ in **b**, and $n = 4$ in **d, f, g**, and **i**, one-way ANOVA). All experiments were repeated at least three times independently.

macrophages[41]. Interestingly, blockade of BMP/SMAD/ RUNX2 signaling pathway significantly increased the intracellular survival of Mtb in macrophages (Fig. 6h), hinting a protective role of this signaling cascade for the host defense against Mtb. We thus further investigated the effects of BMP/SMAD/RUNX2 pathway on host anti-Mtb immune responses, and found that blockade of this pathway in Mtb-infected macrophages remarkably reduced the production of TNF and its responsive molecule, inducible NO synthase (iNOS), both of which play key roles in host anti-Mtb immunity[42,43] (Fig. 6i, j). In addition, inhibition of BMP/SMAD/RUNX2 pathway slightly increased IL-1β and IL-6, and decreased IL-10 production in Mtb-infected macrophages (Supplementary Fig. 10). We then tested whether Mtb-infected macrophages could affect the osteogenic differentiation of MSCs, which play an essential role in heterotopic ossification and could be recruited to TB granulomas[44]. Using macrophage and MSC markers for histological analysis of TB lung sections, we found that a considerable number of host macrophages and MSCs were aggregated together surrounding the granuloma, implying a potential interaction between these two types of cells (Fig. 6k). To confirm the crosstalk between macrophages and MSCs during Mtb infection, we obtained both bone marrow-derived macrophages (BMDMs) and mesenchymal stem cells (BMSCs) from mice for Mtb infection. Consistently, infection with Mtb activated BMP/SMAD/RUNX2 signaling pathway in BMDMs, which effect could be inhibited by LDN-193189 (Supplementary Fig. 11). Alizarin Red S staining demonstrated that BMSCs cultured with conditional media (CM) derived from Mtb-infected BMDMs promoted the formation of BMSC matrix mineralization, as compared to that cultured with control media, CM from BMDMs or CM from Mtb-infected BMDMs with treatment of LDN-193189 (Fig. 6l). Consistently, qPCR analysis showed that BMSCs cultured with CM from Mtb-infected BMDMs exhibited increased Alp, Bglap and Runx2 mRNA levels, as compared to the other treatment groups (Fig. 6m–o). Together, these results suggest that in response to Mtb infection, BMP/SMAD/RUNX2 pathway is activated in host macrophages, which facilitates the production of TNF and iNOS against mycobacterial survival, and also promotes the osteogenic differentiation of MSCs.

Collectively, this study unravels both the pathogenic links and differences among TB, LUAD, and sarcoidosis, providing new insights into TB pathogenesis and suggesting potential molecular markers for differential diagnosis of those clinically important pulmonary diseases (Supplementary Fig. 12).

## Discussion
In this study, we compared lung molecular signature of pulmonary TB with that of LUAD and sarcoidosis, which two diseases have long been linked to Mtb infection[4,6,11]. Based on similar transcriptional changes, we found that lung matrix

remodeling with fibrillar collagen deposition was the most common pathogenic characteristic shared by TB, LUAD, and sarcoidosis. Both type I and III collagens, which are core components of lung interstitial ECM and predominant in chronic fibrotic diseases[25], were increased in lung lesions of TB, LUAD, and sarcoidosis patients, consistent with clinical observation that these diseases are generally associated with fibrotic progression and thus impaired lung architecture and function[2,25,45]. Furthermore, ECM components such as collagens play a crucial role in facilitating tumor cell metastasis and regulating tissue inflammation and autoimmunity[25]. It should be noted that apart from TB, LUAD, and sarcoidosis, some other lung diseases such as chronic obstructive pulmonary disease, idiopathic pulmonary fibrosis, and asthma may also exhibit disordered lung matrix organization[46], and the underlying mechanistic links among those diseases warrant further investigation.

The causal link between TB infection and tumorigenesis has only been partially elucidated[47,48]. Here we found that *MKI67*, a nuclear protein regulating the expression of numerous genes to drive tumourigenesis[49,50], was over-expressed in both TB and LUAD lungs. We then further demonstrated that PtpA contributes to *MKI67*-mediated tumor cell proliferation, migration, and invasion during Mtb infection. These results are consistent with our previous finding that PtpA can enter into the host cell nucleus to regulate nuclear genes to enhance tumor development, and *MKI67* is a potential host target of Mtb effector PtpA[27]. It should be pointed out that other than *MKI67*, there might be additional host genes involved in Mtb-promoted tumor cell proliferation, migration and invasion. Also, besides PtpA, there might be additional Mtb effector proteins that target *MKI67* to regulate these processes. Furthermore, we noticed that Mtb could be carried by infected tumor cells to move through the Matrigel in migration and invasion assays. This might be a potential mechanism employed by Mtb to disseminate more efficiently within lung cancer patients, which notion could be supported by the observation that Mtb can persist in human LUAD-derived epithelial-like cells and spread within the host by coopting host migrating dendritic cells and macrophage cells[51–53].

Previous studies showed a similar blood signature dominated by interferon-signaling-related inflammatory pathways between TB and sarcoidosis patients[21–23], but the similarities of lung signature between two diseases have not been addressed. Among 20 similar signature genes shared by TB and sarcoidosis identified through RNA-Seq analysis, we further confirmed similar over-expression of *MMP12*, *ADAMDEC1*, *CCL19*, *CXCL13*, and *CYP27B1* in TB and sarcoidosis lungs. *MMP12* and *ADAMDEC1* are two previously reported potential pathogenic mediators of lung damage and remodeling most highly over-expressed in sarcoidosis patients[24]. *CCL19* and *CXCL13* genes encode homeostatic chemokines that are inducibly expressed in the lung

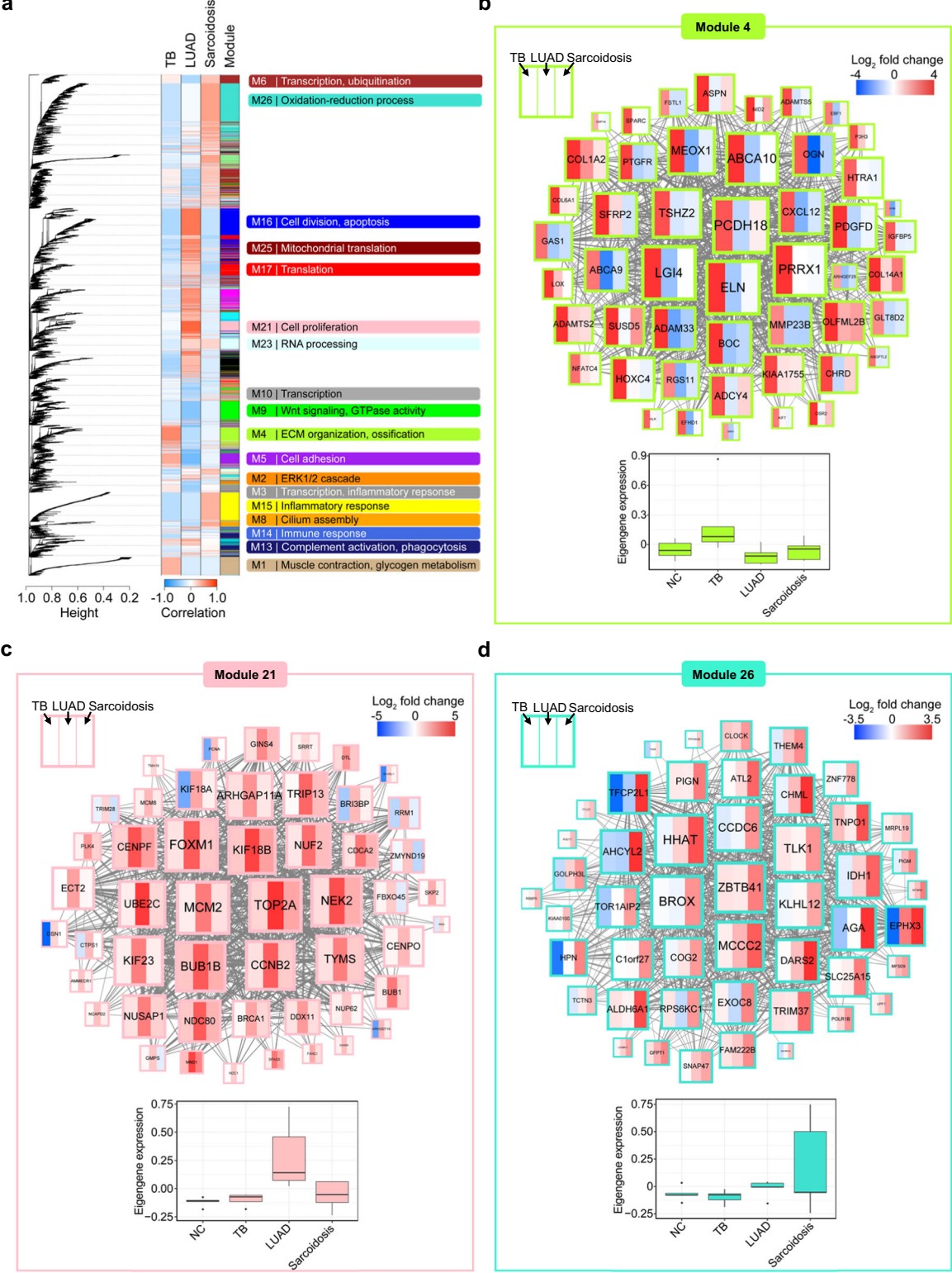

**Fig. 4 Modular transcriptional signatures reveal pathogenic differences among TB, LAUD, and sarcoidosis. a** WGCNA cluster dendrogram based on gene expression profiles of NC, TB, LUAD, and sarcoidosis lung-tissues groups genes ($n = 16,298$) into distinct modules. Each leaf in the tree represents one gene, and the major branches constitute 27 modules (M1-27) labeled by different colors (column 4). The correlations of the corresponding module genes and the studied traits (TB, LUAD, and sarcoidosis) are shown (column 1–3). The modules significantly enriched for gene ontologies associated with certain biological processes (Fisher's exact FDR ≤ 0.05) are indicated on the right. **b**–**d** Gene networks depicting the top 50 highly connected module members (hub genes) for TB-correlated gene module 4 (**b**), LUAD- correlated gene module 21 (**c**) or sarcoidosis-correlated gene module 26 (**d**) in WGCNA. Each gene is shown as a square node with three partitions representing log2 fold-change for TB, LUAD, and sarcoidosis as compared to NC. The lines represent correlation between the gene expression profiles of the two respective genes. The box plots of eigengene expression of each module are shown below the gene network.

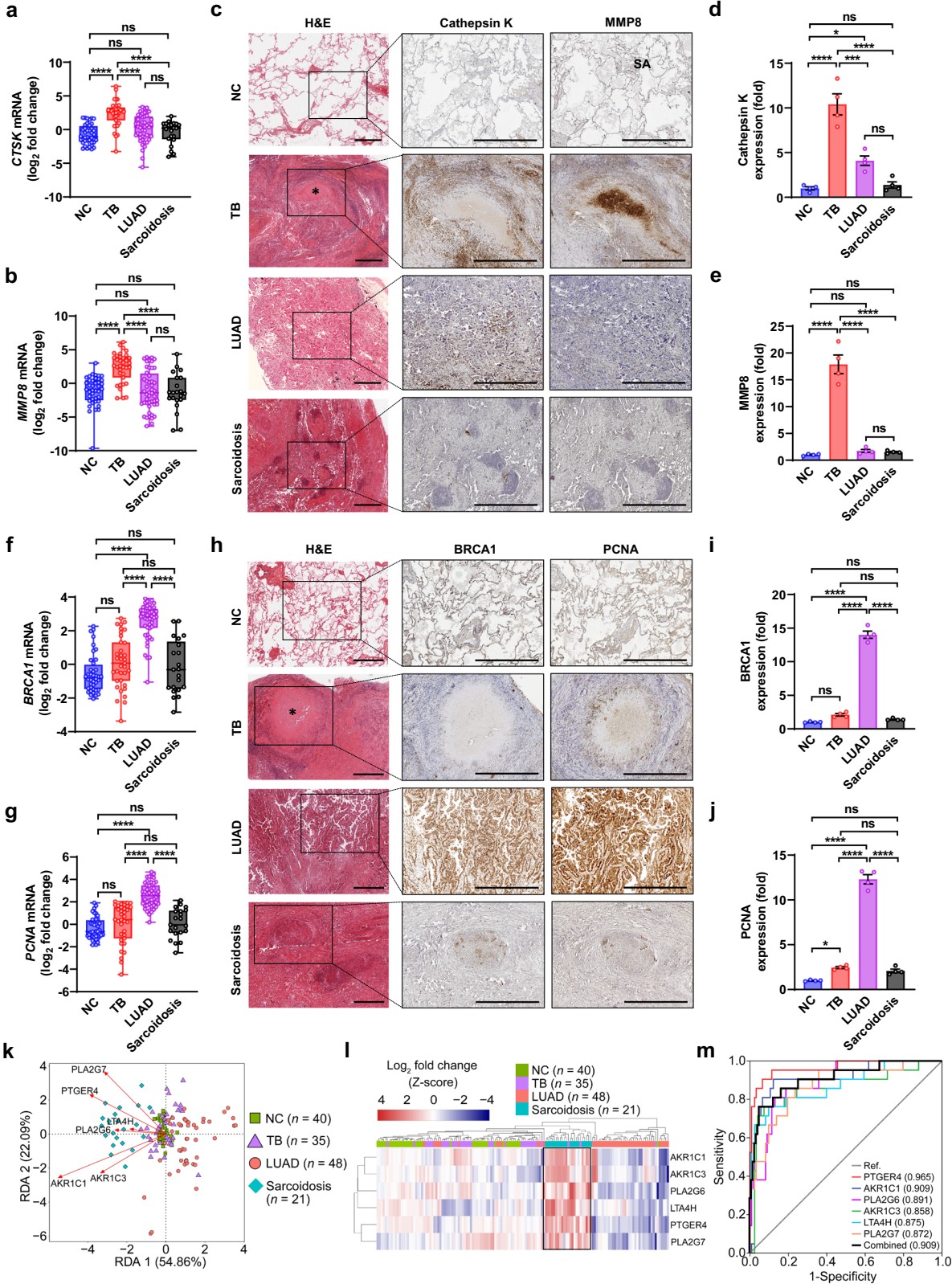

with inflammatory stimuli or pathogen challenges like Mtb infection to shape local immunity[54]. Over-representation of these two molecules indicates similar hyper-inflammatory responses in the lung of TB and sarcoidosis patients. *CYP27B1* gene-encoded CYP27B1 is a key hydroxylase that catalyzes the synthesis of 1,25

$(OH)_2D_3$, the active form of vitamin D which regulates broad immune responses via binding to the vitamin D receptor[55]. The observation of increased *CYP27B1* mRNA in both TB and sarcoidosis lungs is consistent with clinical findings that 1,25 $(OH)_2D_3$ is commonly over-produced in individuals with

**Fig. 5 Specific pathogenic markers for distinguishing TB, LUAD and sarcoidosis patients. a, b** Quantitative PCR analysis of *CTSK* (**a**) and *MMP8* (**b**) mRNAs in each group as indicated. **c** Representative images of histologic and immunohistochemical analysis (for Cathepsin K and MMP8) of NC, TB, LUAD and sarcoidosis lung sections (*n* = 8, 10, 12 and 8, respectively). **d, e** Quantitative analysis of Cathepsin K (**d**) and MMP8 (**e**) immunostaining as in **c**. **f, g** Quantitative PCR analysis of *BRCA1* (**f**) and *PCNA* (**g**) mRNAs in each group as indicated. **h** Representative images of histologic and immunohistochemical analysis (for BRCA1 and PCNA) of NC, TB, LUAD and sarcoidosis lung sections (*n* = 8, 10, 12, and 8, respectively). **i, j** Quantitative analysis of BRCA1 (**i**) or PCNA (**j**) immunostaining as in **h**. **k** Redundancy analysis demonstrating the correlations of *PLA2G6*, *PLA2G7*, *AKR1C1*, *AKR1C3*, *LTA4H* and *PTGER4* with each of the indicated groups. The direction of arrows indicates the positive correlation between gene expression level and samples, and the length of arrows indicates the intensity of the correlation. **l** Heatmap depicting arachidonic acid metabolism-related six-gene signatures of NC, TB, LUAD and sarcoidosis groups according to qPCR results. The box highlights the cluster of sarcoidosis samples. **m** Receiver operating characteristic (ROC) analysis for evaluating the diagnostic potential of the above-mentioned 6 genes for sarcoidosis. The area under the ROC curve (AUC) in parentheses represents the accuracy of the individual and combined genes for distinguishing sarcoidosis samples from the other samples. For **a**, **b**, **f**, and **g**, Box-whisker plot indicates the interquartile range (box), the median value (line within the box) and the maximum and minimum value (whiskers). $P > 0.05$, not significant (ns); ****$P < 0.0001$ (mean ± s.e.m. of $n = 40, 35, 48$, and 21 in NC, TB, LUAD and sarcoidosis groups, respectively, one-way ANOVA). Results are representatives of three independent experiments. For **d**, **e**, **i**, and **j**, four independent visual fields were examined for quantitative analysis. $P > 0.05$, not significant (ns); *$P < 0.05$; ***$P < 0.001$; ****$P < 0.0001$ (mean ± s.e.m. of $n = 4$, one-way ANOVA).

granulomatous inflammation, including both TB and sarcoidosis patients[56]. Collectively, identification of these potential pathogenic mediators shared by TB and sarcoidosis provides clues for their underlying pathogenic links.

Pulmonary TB patients may share similar symptoms and signs with lung cancer or sarcoidosis patients[4,6,16]. In some cases, imaging examinations could not discriminate between TB and cancer[15,57], and the histological appearances of TB and sarcoidosis may be indistinguishable[6,11,16]. Also, false-negative culturing results of Mtb may further challenge the differential diagnosis between TB and cancer[13,15]. According to a report, TB and malignancy can be mistaken for each other at first clinical presentation in nearly one-third of the reporting cases[14]. Another study in a hospital of TB-endemic region reported that a total of 14 out of 70 (20%) "TB patients" with little response to anti-TB treatment received misdiagnosis and were actually lung cancers[13]. Moreover, it was speculated that ~15–20% of the patients might be misdiagnosed if using immunological tests for differential diagnosis of TB and sarcoidosis[58]. Thus, lung tissue-based molecular markers might be helpful for differential diagnosis of these lung diseases in certain cases. Using a modular transcriptional approach, we unraveled different lung characteristics of TB, LUAD, and sarcoidosis. We identified MMP8, also known as neutrophil collagenase, as a TB-specific ECM protease. Intriguingly, MMP8 is massively aggregated in the necrotizing foci of TB, consistent with previous findings that MMP8 mediates Mtb-caused pulmonary matrix destruction and cavitation[59]. MMP8 was shown to be closely associated with neutrophil activation and infiltration in TB lungs[30,59], and TB blood transcriptome exhibits a neutrophil-driven interferon-inducible signature[17]. Thus, our findings support the notion that neutrophils play a predominant role in TB-related inflammation and tissue damage. We also identified BRCA1 and PCNA to be specifically over-represented in LUAD lung lesions and markedly correlated with lung LUAD patient survival. Both BRCA1 and PCNA are critical regulators for DNA replication and repair, whose potential to act as prognostic indicators in lung cancers have been well-documented[60,61]. Lung sarcoidosis modular signature showed over-representation of genes involved in AA metabolic pathway, from which we have identified 6-gene signature that could be used to distinguish sarcoidosis from TB and LUAD. AA metabolism generates diverse intermediates, such as prostaglandins, leukotrienes, and lipoxins, which play critical roles in local immune homeostasis and thus are associated with multiple inflammatory diseases including TB, sarcoidosis and diverse types of cancers[31,62,63]. Therefore, distinct transcriptional signatures in AA metabolic

pathway of these diseases imply different lung immunometabolic status of TB, LUAD and sarcoidosis patients.

Mtb infection has long been linked to lung calcification and ossification without clear molecular mechanisms[34,35]. TB lung signature indicated over-represented genes involved in ossification. As compared to noncalcified TB patients, calcified TB patients exhibited higher expression of osteogenic marker genes in the lungs, indicating a correlation between ossification process and TB lung calcification. Furthermore, we identified the activation of ossification-related BMP/SMAD/RUNX2 signaling pathway and observed the formation of bone spicule with the deposits of calcium in TB lungs suggestive of calcification based on CT findings. We also confirmed that upon Mtb infection, BMP/SMAD/RUNX2 pathway is activated in host macrophages, which facilitates the production of TNF and iNOS against mycobacterial survival and promotes the osteogenic differentiation of MSCs. BMP signaling has been reported to be Mtb-responsive in macrophages[64], which signaling pathway plays a critical role in macrophage-MSC interaction during the ossification process[65,66]. Moreover, the critical roles of TNF and iNOS in host protective immunity against Mtb have been well-documented[42,43]. Therefore, the discovery of the key regulatory role of BMP/SMAD/RUNX2 pathway in TB pathogenesis may provide novel therapeutic targets for host-directed TB treatment. Furthermore, these findings suggest a previously undescribed cooperative role between macrophage and MSC in TB-caused lung ossification, which facilitates host controlling of Mtb infection and repairing of damaged tissues. These findings may also explain for the clinical observation that calcified pulmonary nodules are typical CT findings of inactive TB[35], which phenomenon has been generally regarded as an indicator of a benign outcome. Nevertheless, it should be noted that Mtb can persist in and interact with both macrophages and MSCs in TB granulomas to subvert their biological functions[44,67]. Thus, the calcification progress and disease outcome are determined by the intimate and complex interactions between Mtb and the host.

Our work has some limitations. First, since lung biopsy tissues from healthy people were not available, we thus used disease-uninvolved lung tissues from patients as controls. Second, some of the included TB patients had failed anti-TB drug therapy before they chose surgery treatment, and tissue samples obtained from those patients could not exclude the potential effects of TB drug treatment on TB lung microenvironment. Last, the peak incidence ages of pulmonary TB and sarcoidosis are relatively younger than that of lung cancers[45,68,69], thus the ages of the included TB, LUAD, and sarcoidosis patients (37.4 ± 12.0, 60.4 ±

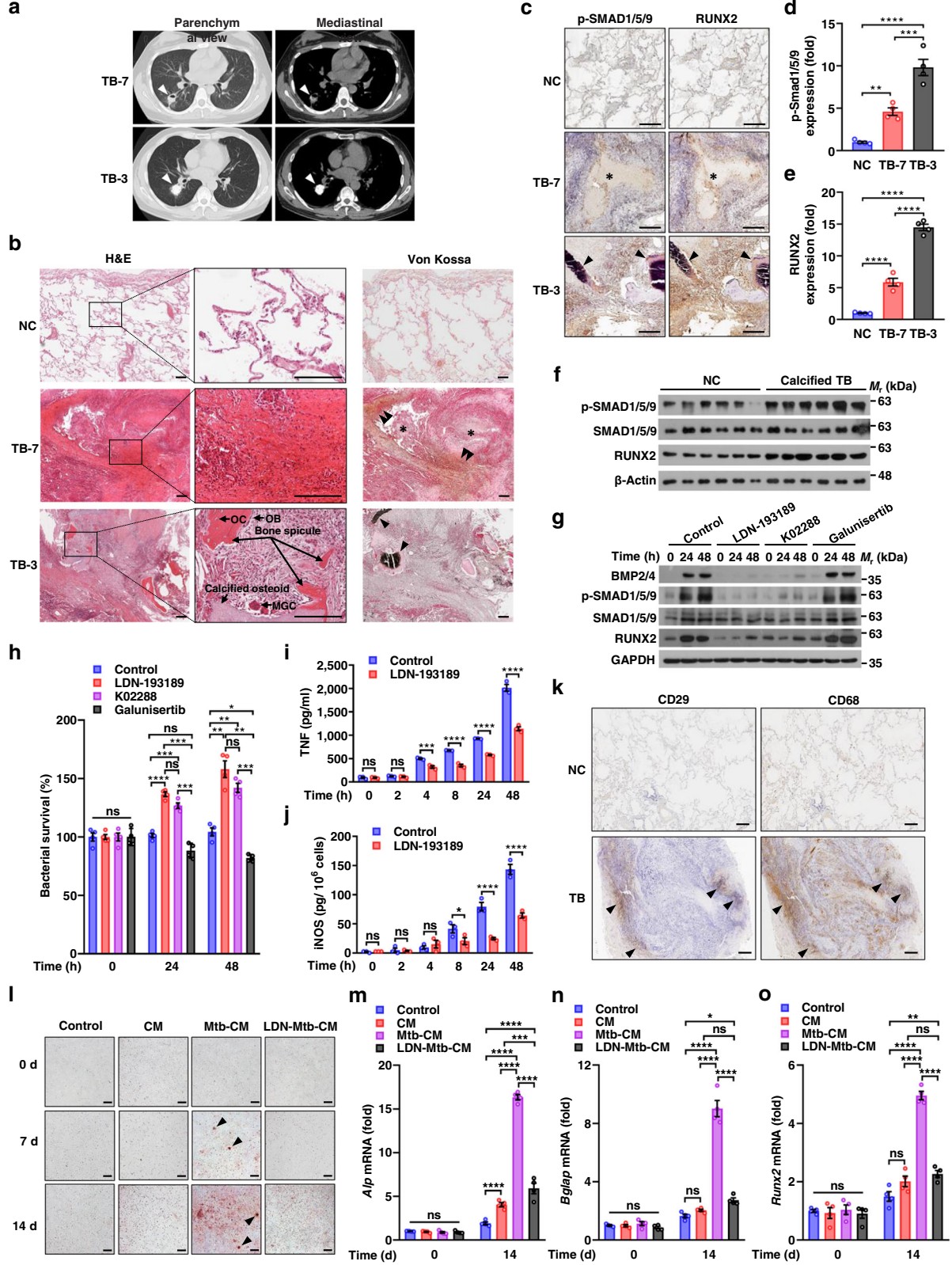

10.4 and 36.4 ± 13.4, respectively) in this study were not well matched.

In summary, this study presents a comprehensive comparison of pulmonary TB lung molecular signature with that of LUAD and sarcoidosis, revealing the potential pathogenic links and differences among these diseases. Our findings also provide new insights into Mtb pathogenesis and potential molecular markers for differential diagnosis of TB, LUAD, and sarcoidosis.

## Methods

**Patient recruitment and tissue specimen collection.** Ethical permission for this study was obtained from the ethics committee of Beijing Chest Hospital and

**Fig. 6 Macrophages control Mtb survival and promote osteogenic differentiation of MSCs via BMP/SMAD/RUNX2 pathway. a** Lung CT scan from patients TB-7 and TB-3. Arrows indicate calcific densities in pathologic lesions. **b, c** Representative images of histologic analysis and Von Kossa staining (**b**), and p-SMAD1/5/8 and RUNX2 immunohistochemical analysis (**c**) of NC and TB lung sections from TB patients as in **a**. Single arrowheads, the osteoid with mass deposits of calcium (black); double arrowheads, dispersed deposits of calcium (brown); asterisks, the necrotizing foci. OC osteocyte; OB osteoblast; MGC multinucleated giant cell. Scale bars, 200 µm. **d, e** Quantitative analysis of p-SMAD1/5/8 (**d**) and RUNX2 (**e**) immunostaining as in **c**. **f** Immunoblot analysis of p-SMAD1/5/9, SMAD1/5/9, RUNX2, and β-Actin in six independent NC or calcified TB lung samples. **g** Immunoblot analysis of BMP2/4, p-SMAD1/5/9, SMAD1/5/9, RUNX2, and GAPDH in U937 cells. Cells were treated with 500 nM LDN-193189, 1 µM K02288, 10 µM Galunisertib or control DMSO, and infected with Mtb for 0–48 h. **h** Survival of Mtb in U937 cells treated as in **g**. **i, j** ELISA of TNF (**i**) and iNOS (**j**) from U937 cells treated with LDN-193189 or control DMSO and infected with Mtb as in **g**. **k** Representative images of CD29 and CD68 immunohistochemical analysis of NC and TB lung sections. Arrowheads indicate the gathering of CD29+ cells (indicating MSCs) and CD68+ cells (indicating macrophages). Scale bars, 200 µm. **l** Representative images of Alizarin Red S staining of BMSCs. BMSCs were cultured with conditional media derived from BMDMs (CM), Mtb-infected BMDMs (Mtb-CM), LDN-193189-pretreated and Mtb-infected BMDMs (LDN-Mtb-CM) or control media for 0–14 d. Arrowheads, the mineralized nodules. Scale bars, 200 µm. **m– o** Quantitative PCR analysis of *Alp* (**m**), *Bglap* (**n**), and *Runx2* (**o**) mRNAs in BMSCs treated as in **l**. $P > 0.05$, not significant (ns); *$P < 0.05$; **$P < 0.01$; ***$P < 0.001$; ****$P < 0.0001$ (mean ± s.e.m. of $n = 4$ in **d**, **e**, **h**, and **m–o**, and $n = 3$ in **i**, **j**, two-way ANOVA). Results are representatives from at least three independent experiments.

Capital Medical University, Beijing, China. Patients were recruited between August 2017 and April 2019, from Beijing Chest Hospital. All study participants were older than 16 years, human immunodeficiency virus (HIV)-negative, and gave written informed consent. In pulmonary TB group, 20 (57.1%) patients were diagnosed by culture-confirmed Mtb in sputum or bronchoalveolar lavage, who received surgery because of little response to anti-TB antibiotic treatments. The other 15 (42.9%) Mtb culture-negative patients were diagnosed by CT findings and histological evidence suggestive of pulmonary TB, and were further confirmed by Xpert MTB/ RIF or PCR assay, and they had not received any medications before diagnostic biopsy or surgical therapy. All included TB patients were confirmed without any other pathogen co-infection or cancers in the lung. In LUAD group, patients were diagnosed by a lung cancer specialist based on histological and radiological analysis in accordance with the standard WHO criteria[70], and were confirmed to be free of any type of infection in the lung. The pathological stage of each patient was verified by experienced pathologists according to the 8th edition of the American Joint Commission on Cancer TNM staging system[71]. All included LUAD patients had not received any radiation or chemotherapy before surgery. In pulmonary sarcoidosis group, patients were diagnosed by a sarcoidosis specialist based on radiological and histological analysis showing evidence of well-formed, noncaseating granuloma without identifiable infection and atypical pathological features, and they were further classified into different roentgenographic stages of intrathoracic changes[45,72]. None of sarcoidosis patients had received any medications before sampling. All of the disease samples were obtained during the surgical resection or diagnostic biopsy, including 35 TB, 48 LUAD, and 21 sarcoidosis lung-tissue samples (Supplementary Data 1). Disease-free lung tissue samples were excised from normal lung parenchyma adjacent to disease lesions from patients who received pneumonectomy. After being verified to have normal lung histology by a certified pathologist, a total of 40 tissue samples (among which 7 samples were from TB patients, 29 were from LUAD patients and 4 were from sarcoidosis patients) were included into NC group for analysis (Supplementary Data 1). All collected tissue samples were preserved in RNA*later* Solution (Thermo Fisher) at −80 °C according to manufacturer's instructions until they were used in the experiments.

**RNA-Seq library preparation and sequencing**. Human lung tissues were used for total RNA extraction with RNeasy Plus Mini Kit (Qiagen) according to the manufacturer's protocol. RNA purity was assessed using a Nanodrop ND-2000 (Thermo Scientific), and each RNA sample had an A260:A280 ratio above 1.8 and an A260:A230 ratio above 2.0. RNA integrity was evaluated using the Agilent 2200 TapeStation (Agilent Technologies), and each sample had the RNA Integrity Number (RIN) above 7.0. The ribosomal RNAs (rRNAs) were removed using Ribo-Zero rRNA Removal Kits (Illumina). RNA libraries were then constructed by NEBNext Ultra RNA Library Prep Kit for Illumina (New England Biolabs) and evaluated using the Agilent 2200 TapeStation and Qubit® 2.0 (Life Technologies). Sequencing was performed by RiboBio Co., Ltd. on an Illumina HiSeq 3000 machine with paired-end 150-bp reads.

**RNA-Seq data analysis**. The adaptors and low-quality bases assessed using FASTQC 0.11.3 were trimmed by Trimmomatic 0.39 using the following options: TRAILING:20, MINLEN:235, and CROP:235. Trimmed reads were then aligned to the ensembl 79 (GRCh38.p2) reference genome using STAR 2.4.2a (ref. [73]). FeatureCounts 1.6.2 was subsequently employed to convert aligned short reads into read counts for each sample[74]. Genes with less than ten counts in two or more samples were removed. The data were then analyzed using R 3.4.4 and DEseq2 1.18.1 (ref. [75]). Differentially expressed genes of each disease group were identified using Wald statistics test, with fold-change >2 and Benjamini–Hochberg (BH)-

adjusted $P < 0.1$ as compared to NC group. Cluster analysis and visualization was performed using the pheatmap Bioconductor package 1.0.12.4.

**TCGA data analysis**. LUAD gene expression profiles were downloaded from TCGA using GDC Data Transfer tool 1.3.0. The data included clinical information and gene expression values of 510 LUAD lung tissues and 58 normal lung tissues. Differentially expressed genes were identified using the R package DESeq2 1.18.1 (ref. [75]), with fold-change >2 and BH-adjusted $P < 0.1$. For overall survival analysis, higher and lower mRNA levels of a given gene were divided at the median value based on 510 tumor samples. Hazard ratios, log-rank $P$-values and Kaplan–Meier survival curves were calculated using survival package 3.1.8. Log-rank $P$-values < 0.05 were considered to be significant.

**Microarray data**. Represented lung-tissue transcriptional profiles and clinical annotations of pulmonary sarcoidosis ($n = 6$) and the matched control ($n = 6$) were downloaded from the National Center for Biotechnology Information Gene Expression Omnibus (http://www.ncbi.nlm.nih.gov/geo, GSE16538). The dataset was processed and normalized using Robust Multi-array Average (RMA) algorithm[76]. Differentially expressed genes were identified using the limma package 3.34.9[77], with fold-change >2 and BH-adjusted $P < 0.1$.

**Enrichment analysis**. Gene ontology (GO), Kyoto Encyclopedia of Genes and Genomes (KEGG) and Reactome enrichment analysis were carried out using R clusterProfiler package 3.6.0[78], and the enriched items with Benjamini–Hochberg (BH)-adjusted $P < 0.05$ was considered to be statistically significant. Gene set enrichment analysis (GSEA) was performed using online R code from Molecular Signatures Database (http://software.broadinstitute.org/gsea/msigdb/index.jsp). GSEA calculates the enrichment score (ES) by applying weighted Kolmogorov-Smirnov statistic to a running sum of the ranked list with 1000 permutations. The ES was then normalized to account for the size of the inputted gene set. The false discovery rate (FDR) < 0.05 were assumed to be statistically significant. For the protein–protein interaction network analysis, the Search Tool for the Retrieval of Interacting Genes/Proteins (STRING) database 11.0 was employed[79].

**Weighted gene co-expression network analysis**. The R package weighted gene co-expression network analysis (WGCNA) 1.68 was used to construct a co-expression network using reads per kilobase of transcripts per million-mapped (RPKM) data from RNA-seq[29]. A thresholding power of 8 was chosen (as it was the smallest threshold that resulted in a scale-free $R^2$ fit of 0.8) and the network was created by calculating the component-wise minimum values for topologic overlap (TO). Using 1-TO (dissTOM) as the distance measure, genes were hierarchically clustered in a dendrogram. Initial module assignments were determined by using a dynamic tree-cutting algorithm (cutreeHybrid, using default parameters). The resulting 27 modules of co-expressed genes were used to calculate module eigengenes (MEs). MEs were correlated with different biological traits indicating certain TB-, LUAD- or sarcoidosis-specific modules. Top 50 hub genes with high intra-modular connectivity and high correlation were calculated and exported into Cytoscape 3.6.0 to create interaction networks. The FDR-corrected $P$-values from multiple comparisons across modules were reported.

**Redundancy analysis and receiver operating characteristic curve analysis**. The top-6 most important arachidonic acid metabolism-related genes correlated with sarcoidosis disease were determined by Gini score of each gene calculated from log₂-transformed quantitative PCR (qPCR) expression values, using the R package of randomForest 4.6.14. The R package of vegan 2.5.6 was used for redundancy analysis to depict the relationships between the expression values of selected genes

and samples from NC, TB, LUAD, and sarcoidosis groups ($n$ = 40, 35, 48, and 21, respectively). For evaluation of diagnostic potential of selected arachidonic acid metabolism-related genes for sarcoidosis, receiver operating characteristic curves were generated based on $\log_2$-transformed quantitative qPCR expression values using IBM SPSS Statistics 22.0, followed by Delong test. The sum of $\log_2$-transformed expression values of selected genes was used to evaluate the combined diagnostic potential.

**Antibodies and reagents.** Rabbit anti-PtpA antibody was produced and purified as described previously[80]. Briefly, GST-tagged PtpA protein was purified and solubilized in Freund's complete adjuvant for injection into rabbits. The antibody specific to PtpA was isolated by passaging the immunized rabbit serum on protein A agarose (#sc-2001; Santa Cruz). The following commercially available antibodies were used in this study: anti-collagen I (#NB600-408, Novus Biologicals, 1:400 for immunohistochemical staining), anti-collagen III (#ab2345, Abcam, 1:200 for immunohistochemical staining), anti-Ki-67 (#NB500-170, Novus Biologicals, 1:100 for immunohistochemical staining and 1:2000 for immunoblot analysis), anti-Cathepsin K (#sc-48353, Santa Cruz, 1:200 for immunohistochemical staining), anti-MMP8 (#sc-514803, Santa Cruz, 1:100 for immunohistochemical staining), anti-BRCA1 (#ab16780, Abcam, 1:100 for immunohistochemical staining), anti-PCNA (#sc-25280, Santa Cruz, 1:100 for immunohistochemical staining), anti-p-SMAD1/5/9 (#13820, Cell Signaling Technology, 1:100 for immunohistochemical staining and 1:1000 for immunoblot analysis), anti-SMAD1/5/9 (#ab66737, Abcam, 1:1000 for immunoblot analysis), anti-RUNX2 (#sc-101145, Santa Cruz, 1:100 for immunohistochemical staining and 1:2000 for immunoblot analysis), anti-BMP2/4 (#sc-137087, Santa Cruz, 1:1000 for immunoblot analysis), anti-GAPDH (#sc-25778, Santa Cruz, 1:4000 for immunoblot analysis), anti-β-actin (#A2228, Sigma-Aldrich, 1:4000 for immunoblot analysis), anti-Tubulin (#T5168; Sigma-Aldrich), Anti-CD29 (#sc-9970, Santa Cruz, 1:100 for immunohistochemical staining) and anti-CD68 (#sc-17832, Santa Cruz, 1:100 for immunohistochemical staining). LDN-193189 (#S2618), K02288 (#S7359) and Galunisertib (#S2230) were purchased from Selleck.

**Bacterial strains.** *M. tuberculosis* (Mtb) H37Rv strains (Catalog No. 27294, ATCC) were grown in Middlebrook 7H9 broth (7H9) supplemented with 10% oleic acid-albumin-dextrose-catalase (OADC) and 0.05% Tween-80 (Sigma), or on Middlebrook 7H10 agar (BD) supplemented with 10% OADC. The pJV53 system was used to create Mtb H37Rv strain with deletion of the gene encoding PtpA (Mtb ΔPtpA)[81], and pMV306 plasmid (provided by W. R. Jacobs, Albert Einstein College of Medicine, Yeshiva University) was used to complement the strain Mtb ΔPtpA with WT PtpA (Mtb ΔPtpA:PtpA). The oligonucleotides and plasmids used for mycobacterial recombination and complement were described previously[80].

**Mammalian cell lines.** A549 cells (ATCC CCL-185) and U937 cells (ATCC CRL-1593.2) were obtained from the American type culture collection (ATCC). A549 cells were cultured in Dulbecco's modified Eagle's medium (DMEM; Gibco) with 10% fetal bovine serum (FBS; Hyclone), and U937 cells were maintained in RPMI 1640 medium (Gibco) with 10% FBS.

**Preparation of bone marrow-derived macrophages and mesenchymal stem cells from mice.** Mice on C57BL/6 genetic background were housed in a specific pathogen-free (SPF) facility on the basis of standard humane animal husbandry protocols, which were approved by the animal care and use committee of the Institute of Microbiology (Chinese Academy of Sciences). Bone marrow-derived macrophages (BMDMs) were collected from tibiae and femurs of 7-8 weeks old mice. After lysis of red blood cells, BMDMs were cultured in DMEM supplemented with 10% FBS, 1% Penicillin-Streptomycin Solution (Caisson) and Murine M-CSF (Pepro Tech) for 4–6 days. Bone marrow mesenchymal stem cells (BMSCs) were also collected from tibiae and femurs of 7-8 weeks old mice and cultured following a published protocol[82], and were used for experiments at the fourth passage.

**Transfection.** siRNAs for Ki-67 (5′-GGTCACACTGAGGAATCAA-3′) and the scrambled negative control (NC) siRNAs were synthesized and purified from GenePharma, referring to Vanneste et al.[83]. A549 cells were seeded in six-well plates at a density of $5.0 \times 10^5$ cells per well, and the transfection was carried out with 100 pmol of siRNAs per well combined with Lipofectamine 2000 (Invitrogen) according to the manufacturer's instructions.

**Macrophage infection and colony-forming unit counting.** U937 cells or BMDMs were seeded in six-well plates at a density of $1.0 \times 10^6$ cells per well. For infection of U937 cells, U397 cells were pretreated with 10 ng/ml phorbol 12-myristate 13-acetate (PMA, Sigma) overnight to differentiate into adherent macrophage-like cells. Mycobacterial strains were prepared based on a standard experimental procedure[84]. Basically, bacteria were grown to mid-logarithmic phase for infection experiments, and some of the log-phase cultures were flash frozen and stored at −80 °C for further repeated experiments. Before infection, bacteria were washed twice in PBS, and then pelleted and thoroughly resuspended using the cell culture medium with 0.05% Tween-80. Thereafter, cells were infected with Mtb strains at a

multiplicity of infection (MOI) of 1 for U937 cells and BMDMs. The media were then discarded, following by three washes with phosphate buffered saline (PBS) buffer, and the cells were incubated again with the fresh medium and cultured until the designated time points. For bacterial colony-forming unit (CFU) counting, cells were washed thrice with PBS buffer and lysed in 7H9 broth with 0.05% SDS at each time point. Several sets of serially gradient dilution of the lysates were prepared in 7H9 broth and then cultivated on 7H10 agar plates. The colonies were counted after 3–4 weeks.

**Cell proliferation assays.** For Cell Counting Kit-8 (CCK-8) assay, A549 cells were transfected with Ki-67 siRNA or NC siRNA for 24 h, followed with or without infection of Mtb stains at a MOI of 5 for 1 h. Thereafter, cells were extensively washed with PBS buffer, and suspended in DMEM with 10% FBS following a short digestion using 0.25% Trypsin-EDTA (Gibco). Cells were then seeded at a density of 2000 cells per well in 96-well plates and incubated at 37 °C. At each of designated time points, an aliquot of 10 μl of CCK-8 solution (DOJINDO) was added into the wells and incubated for 1 h. Thereafter, the absorbance was measured at 450 nm to determine the viable cells in each well. The experiment was performed in triplicates and repeated three times independently. For 5-ethynyl-2′-deoxyuridine (EdU) cell proliferation assay, cells were seeded on poly-lysine-coated coverslips and then transfected with siRNAs and infected with Mtb stains as described above. At 24 h post-infection, cells were treated with EdU at concentration of 10 μM for 3 h, and then fixed and permeabilization for incorporated EdU detection using Beyo-Click™ EdU-594 Cell Proliferation Assay Kit (Beyotime) according to the manufacturer's instructions. Nuclei were stained with Hoechst 33342 (Beyotime). Confocal images were taken with Olympus FV1000 confocal microscope and analyzed by FV10-ASW 3.0 software. Five independent visual fields were examined for quantitation of EdU-positive cells. The assay was repeated three times independently.

**Transwell migration and invasion assays.** For migration assay, A549 cells were transfected with NC or *MKI67* siRNA for 24 h, followed by incubation with Mtb strains at a MOI of 5 for 1 h. Cells were then seeded onto the apical side of the chamber of Cell Culture Inserts (24-well format, 8.0-μm pore size, Falcon) at a density of $3.0 \times 10^4$ cells in serum-free DMEM. DMEM containing 20% FBS was added to the basal compartment to serve as a chemoattractant, and cells were allowed to migrate for 12 h. Thereafter, the cells on the apical side of the chamber were gently scraped off using wetted cotton swabs, and the cells migrated to the basal side were fixed in 95% ethanol and stained for 30 min in a 0.1% Crystal Violet solution in 1x PBS buffer. For invasion assay, the chambers were pre-coated with 100 μl of 2 mg/ml Matrigel (BD Biosciences), and the other procedures were carried out as described above. For immunofluorescence microscopy, Alexa Fluor 488 succinimidyl ester (Invitrogen) was used for bacteria staining before infection as described before[85]. The cells migrated to the basal side were fixed in 4% paraformaldehyde (PFA) for 15 min, and then washed twice with PBS buffer, followed by permeabilization with 0.5% Triton X-100 for 5 min. After two washes, each chamber membrane was carefully separated using a scalpel and moved onto a glass slide. The coverslips were then mounted onto glass slides using DAPI Staining Solution (Beyotime). Confocal images were taken with Olympus FV1000 confocal microscope and analyzed by FV10-ASW 3.0 software. Four independent visual fields were examined to count the number of cells that had moved to the bottom chamber. The assays were repeated three times independently.

**qPCR analysis.** Total RNA extraction of A549 cells, BMSCs, and human lung tissues were carried out using RNeasy Plus Mini Kit (Qiagen) according to manufacturer's protocols. The reverse-transcription of RNA was accomplished by using a 1st Strand cDNA Synthesis SuperMix (Yeasen) and performed to qPCR analysis with qPCR SYBR Green Master Mix (Yeasen) on Applied Biosystems 7500 Real-Time PCR system. For A549 cells and BMSCs, quantitative expression of targeted genes was normalized to human *ACTB* and mouse *GAPDH*, respectively. The experiments were performed in quadruplicates and repeated three times independently. For human lung tissues, quantitative expression of targeted genes was normalized to *ACTB* and *18S* to confirm that results were not due to variability within housekeeping gene expression, and the data presented in the figures represent analysis based on normalization to *ACTB*. The experiments were repeated at least two times independently. All qPCR primers used in this study were listed in Supplementary Data 7.

**Histology and immunohistochemical analysis.** The lung tissues were fixed in 10% formalin and embedded in paraffin, and then were cut at 4 μm thickness for preparation of serial section slides. Tissue sections were stained with hematoxylin & eosin (H&E) for histological examination or subjected to immunohistochemical analysis with each of the indicated primary antibodies according to standard techniques. Three independent visual fields were examined for quantitative analysis using ImageJ 1.50e with an IHC Toolbox plugin according to the instructions (https://imagej.nih.gov/ij/plugins/ihc-toolbox/index.html). Trichrome Stain Kit (Abcam) was used for Masson's trichrome stain in line with the manufacturer's protocols. For analysis of calcium deposits, tissue sections were subjected to Von Kossa stain using Von Kossa Stain Kit (Abcam) according to manufacturer's

instructions. The histological and immunohistochemical analysis were performed by two trained technologists and verified by a pathologist.

**ELISA**. U937 cells were treated with 500 ng/ml LDN-193189 or control DMSO and were infected with Mtb as described above. At each time point after infection, $2.0 \times 10^6$ cells were collected and lysed in 200 μl 1x Cell Extraction Buffer PTR for quantitative detection of iNOS by using Human iNOS SimpleStep ELISA Kit (#ab253217, Abcam). Cell culture media were used for quantitative detection of released cytokines including TNF, IL-1β, IL-6, and IL-10 by using ELISA Kits (Human TNF ELISA: #ELH-TNFa-1, RayBiotech; Human IL-1β ELISA: #ELH-IL1b-1, RayBiotech; Human IL-6 ELISA: #ELH-IL6-1, RayBiotech; Human IL-10 ELISA: #ELH-IL10-1, RayBiotech). All experiments were performed in triplicates and repeated three times independently.

**Immunoblot analysis**. Cells were lysed in RIPA Lysis Buffer (Beyotime) supplemented with 1% protease inhibitor cocktail (Bimake). Lung tissues were disrupted and homogenized in the same lysis buffer using a high-throughput tissue homogenizer (Xinyi Co., Ltd.). Proteins were separated by sodium dodecyl sulfate (SDS)-polyacrylamide gel electrophoresis (PAGE) and transferred to polyvinylidene difluoride membranes (Millipore). The membranes were blocked with 5% skimmed milk powder in Tris-buffered saline containing 0.1% Tween-20 (TBST) for 1 h at room temperature (RT) and subsequently incubated with primary antibodies overnight at 4 °C. The membranes were then incubated with goat anti-mouse IgG or goat anti-rabbit IgG conjugated to horseradish peroxidase (HRP) for 1 h at RT after three washes of 10 min each with TBST. Finally, the membranes were developed by Immobilon Western Chemiluminescent HRP Substrate (Millipore) after three washes with TBST again and exposed to x-ray film. All experiments were independently validated with three biological replicates.

**Osteogenic differentiation of BMSCs**. BMDMs were cultured in DMEM containing 10% FBS with treatment of 500 nM LDN-193189 or control DMSO for 2 h and were then infected with or without Mtb H37Rv as described above. BMDMs were further cultured in fresh medium until 24 h post-infection, and the culture supernatants were collected and filtrated through a 0.22-μm filter (Millipore). To investigate the osteogenesis of BMSCs, the culture medium of BMSCs was supplemented with the collected supernatants at a ratio of 1:2 to obtain the conditioned medium (CM) and was then replaced by fresh medium containing the same CM every third day. At each of the designated time points, BMSCs were collected for qPCR analysis as described above, or subjected to alizarin red S stain (ARS). For ARS, cells were washed with PBS buffer twice and fixed in 4% PFA for 10 min. After two washes with deionized water, cells were stained with 40 mM ARS solution (pH 4.2) for 20 min. Cells were then washed three times with deionized water and air dried, and images were taken using a light microscope at ×10 magnification.

**Statistics and reproducibility**. Data are presented as mean and standard error of mean (s.e.m.) or median and 25–75% interquartile range following $\log_2$-based transformation, according to data distribution. One-way or two-way ANOVA analysis followed by multiple comparisons were used for statistical analysis of continuous variables, and the Pearson chi-square test or Fisher exact test were used for categorical variables. The quantified data with statistical analysis were performed using GraphPad Prism 8.0. $P$-values < 0.05 were considered to be statistically significant. Details of statistical analyses of experiments and number of biological replicates ($n$) can be found in the figure legends. Unless otherwise indicated, all experiments were performed at least three times.

**Reporting summary**. Further information on research design is available in the Nature Research Reporting Summary linked to this article.

## Data availability
The source data underlying the main and Supplementary Figures are provided in Supplementary Data 9. The original unprocessed blot images are provided in Supplementary Fig. 13. The RNA sequencing data from this study have been deposited to the NCBI Sequence Read Archive and are accessible through GEO Series accession number GSE148036. All other data that support the findings of this study are available from the corresponding author upon reasonable request.

## Code availability
The source code used in this study were created with R version 3.4.4 and are available on GitHub (https://github.com/LuShuYangMing/RNAseq-Source-Code), where a brief instruction is attached. STRING database (https://string-db.org/) was used for Fig. 1d, f and Supplementary Figs. 5c and 6c, according to online instructions. Cytoscape 3.6.0 software was used for Fig. 4b–d. IBM SPSS Statistics 22.0 was used for Fig. 5m. GSEA source code (http://software.broadinstitute.org/gsea/msigdb/index.jsp) was used for Supplementary Figs. 3b and 9a.

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

## Acknowledgements

This work was supported by the National Key Research and Development Program of China (2017YFA0505900), the National Natural Science Foundation of China (81825014, 31830003), the Strategic Priority Research Program of the Chinese Academy of Sciences (XDB29020000), the National Science and Technology Major Project (2018ZX10101004), and the Key Program of Logistics Research (BWS17J030). We thank Y. Zhou, X. Yang and J. Hao (Core Facility for Protein Research, Institute of Biophysics, Chinese Academy of Sciences, Beijing) for helping with histological and immunohistochemical analysis.

## Author contributions

C.H.L. conceived the project; Y.P. and Z.Liu established the clinical cohorts and collected lung samples; C.H.L., Q.C. and Z.Lu planed the experiments and analyzed the data; Q.C., Z.Lu, Y.Z., F.Z., and C.Q. performed the experiments. B.L., J.W., and L.Z. contributed some critical experimental materials and guidance; Q.C. and C.H.L. wrote the paper, with critical input from all other authors; all authors read and approved the final version of the paper.

## Competing interests

The authors declare no competing interests.
