## [Peer Review File · Communications Biology]

Reviewers' comments:

Reviewer #1 (Remarks to the Author):

This is an interesting study that presents a comprehensive comparison of pulmonary TB lung molecular signature with that of lung adenocarcinoma and sarcoidosis, revealing the potential pathogenic links and differences among these diseases. The main aspects of the study involve the transcriptome and comparison between shared and unique molecular signatures of lung adenocarcinoma, sarcoidosis and pulmonary tuberculosis. They revealed that infection with Mtb induces upregulation of MKI67 mRNA in lung adenocarcinoma cell line and that MKI67 is involved in migration and invasion of A549 cells. They showed that MMP8 was specifically increased in TB lungs as compared to the other groups and verified this finding by qPCR and IHC. They presented that BRCA1 and PCNA was specifically increased in lung adenocarcinoma as compared to the other groups, which was also verified by qPCR and IHC. Moreover, they presented that AA metabolism-related genes (including PLA2G6, PLA2G7, AKR1C1, AKR1C3, LTA4H and PTGER4) were positively correlated to sarcoidosis. Furthermore, they show that Mtb activates BMP/SMAD/RUNX2 signaling pathway and induces pulmonary ossification in TB patients.

Specific comments about manuscript:

1. The study refers to Mtb infection for TB, while for the whole genome analysis only 2 out of 5 had confirmed culture-Mtb assay by Table 1. Mtb confirmation tests performed like Xpert or PCR should be added to Tables 1 and 2. Perhaps by changing to Confirmation of Mtb (culture/Xpert/PCR).
2. Two out of 5 TB analyzed by WGS received treatment before surgery, although the treatment was unsuccessful this is something that should be taken into account when analyzing the micro-environment, which could definitely have an impact on it and should be added to the discussion.
3. Age was not matched in the study and should be added to the discussion since it could have an impact as well.
4. The number of samples used for the validation of expression by qPCR, IHC were not mention for MMP8, BRCA1, PCNA and the AA metabolism related genes.
5. They mention lung adenocarcinoma as a chronic disease, which should be changed to terminal one.
6. There are discrepancies between supplementary Table 1 and Table 2 regarding the NC; numbers in Signs of Calcification; Signs of fibrosis; The tables should be checked.

Overall, the manuscript is interesting and presents mostly original findings which would be of interest to the scientific community.

Reviewer #2 (Remarks to the Author):

This study compared and contrasted TB, sarcoidosis, and adenocarcinoma of the lung by examining the transcriptional signatures and histologic features of tissue samples from patients with these diseases. They found that gene expression related to ECM organization (and particularly collagen matrix remodelling), immune response, cell growth and proliferation were enriched among the 3 disease groups. They further explored matrix remodelling by staining for collagen I and collagen III in tissues samples from each disease group, and found increased expression of collagen types I and III.

My main concerns are that it lacks focus and it doesn't develop a story / explore a concrete hypothesis.

This reads as potentially many papers in 1:

Distinguishing gene signatures across pulmonary diseases (this has been done previously - so would have to have a novel bent, such as their use of lung tissue rather than blood as the tissue sample source)

Ossification in lung disease: mechanistic and morphological features

Pulmonary remodelling in TB, sarcoid and AD

TB infection and tumorigenesis

The background information could be improved:

The authors state that, for AD and sarcoidosis, or for AD and TB, "discriminating these diseases could be challenging". This is incorrect. This has not been reported, and as a respirologist, I have not confused the clinical presentations of AD and sarcoidosis, or AD and TB. The references used to support this claim also do not address diagnostic confusion for AD and these granulomatous diseases.

Method concerns:

The "normal control" tissue is non-affected lung tissue from disease patients. This is not a valid control group, unless they wanted to compare, for each patient, disease and non-diseased lung. Also it wasn't clear how many control samples were derived from each disease category. The

suppl. data does not identify this.

Some of the findings are an overstretch:

"In conclusion, identification of these potential pathogenic mediators of AD or SA shared by TB indicates the potential causal links between TB infection and lung AD or SA." This is unfounded; shared pathways alone does not imply shared aetiology. In the discussion they use the term "pathogenic link" which is more appropriate.

"These findings indicate that Mtb promotes the proliferation, migration and invasion of A549 cells partially depending on MKI67, which mechanism might also benefit its own dissemination within the host." This is a major statement and could be a major finding if they developed a line of investigation around it.

The finding that "all these diseases drove pulmonary remodeling with various degrees of matrix destruction and fibrous protein deposition" is not surprising but has not before been adequately evaluated or definitely established. However, the interpretation that "collagen matrix remodeling in the lung is a common pathogenic characteristic of pulmonary TB, AD and SA." might (and very likely does!) apply more broadly to pulmonary disease. Other disease groups would have been a good control here.

The figures are generally very good. Some were even stunning. But it wasn't always clear why certain data over others was presented. Scale bars are missing for several histology figures. Small point, for reviewers it would be easier to read the figure legends if they were placed near the figure, and if the figures were labeled. The data for the "Transwell migration and invasion assays" in figure 3d is not clearly demonstrated as currently presented.

Minor points:

CXR stages in sarcoidosis are not based on the 1999 statement but rather are Scadding stages. No need to abbreviate sarcoidosis (to SA) - this is not standard in the literature.

AD is not typically described as or considered to be a "widespread chronic disease".

It wasn't clear who read the slides of tissue sections. A pathologist expert in these diseases?

With regards to the H37Rv MTb strains used - it would be good practice to acknowledge the source of the strains.

The validity of using freshly thawed frozen stock to infect cells is controversial (bacteria are not going to be actively replicating and at a great level of fitness).

Generally, infection occurs in exponential phase of growth.

Reviewer #3 (Remarks to the Author):

Overall this is a substantial and well-presented article that provides important insights into the molecular mechanisms underlying the link between lung calcification and ossification in TB, as well as the identification of potential markers to better distinguish between TB, AD and SA.

However, I do not find the initial data (Figs 1 and 2) particularly novel. It is not surprising to me that three diseases well known to cause significant damage to the lung would have similar molecular signatures related to immune response, metabolic processes and pulmonary remodeling, or that signature genes between TB and AD involved in cell growth and proliferation were most directly correlated with tumor development. Similarly the involvement of MKI67 is not surprising in this context.

However, the idea of Mtb directly promoting the proliferation, migration and invasion of human lung adenoma A549 cells is interesting (Fig 3), but the authors use a CCK-8 assay to measure cell proliferation and this only determines cell viability, a proper cell proliferation assay should be performed. This part of the study could also be improved, especially considering the rest of the article is better developed. To add strength to this section it would be beneficial if the authors could show that Mtb-secreted PtpA regulates MKI67 in the A549 cells.

The discussion should include a section “limitations of the study”: potential points to discuss: 1) the caveat/strength of normal controls; they are actually derived from the patients. 2) the authors argue consistently that it is very difficult to differentiate between the three diseases, but they by themselves obviously did not have problems doing so; how high is the likelihood of misdiagnosing?

Point-by-point responses to reviewers' comments

Chai, et al., “Lung molecular signatures reveal pathogenic links and differences among pulmonary tuberculosis, adenocarcinoma and sarcoidosis” (COMMSBIO-20-0930-T).

Reviewers' comments:

Reviewer #1

Remarks to the author:

This is an interesting study that presents a comprehensive comparison of pulmonary TB lung molecular signature with that of lung adenocarcinoma and sarcoidosis, revealing the potential pathogenic links and differences among these diseases. The main aspects of the study involve the transcriptome and comparison between shared and unique molecular signatures of lung adenocarcinoma, sarcoidosis and pulmonary tuberculosis. They revealed that infection with Mtb induces upregulation of MKI67 mRNA in lung adenocarcinoma cell line and that MKI67 is involved in migration and invasion of A549 cells. They showed that MMP8 was specifically increased in TB lungs as compared to the other groups and verified this finding by qPCR and IHC. They presented that BRCA1 and PCNA was specifically increased in lung adenocarcinoma as compared to the other groups, which was also verified by qPCR and IHC. Moreover, they presented that AA metabolism-related genes (including PLA2G6, PLA2G7, AKR1C1, AKR1C3, LTA4H and PTGER4) were positively correlated to sarcoidosis. Furthermore, they show that Mtb activates BMP/SMAD/RUNX2 signaling pathway and induces pulmonary ossification in TB patients.

R: We thank the reviewer for the encouraging comments on our manuscript.

Specific comments about manuscript:

1. The study refers to Mtb infection for TB, while for the whole genome analysis only 2 out of 5 had confirmed culture-Mtb assay by Table 1. Mtb confirmation tests performed like Xpert or PCR should be added to Tables 1 and 2. Perhaps by changing to Confirmation of Mtb (culture/Xpert/PCR).

R: We thank the reviewer for pointing out this issue and for the constructive suggestion. We have added the information of Mtb confirmation tests including culture, Xpert and PCR into Supplementary Tables 1, 2.

2. Two out of 5 TB analyzed by WGS received treatment before surgery, although the treatment was unsuccessful this is something that should be taken into account when analyzing the micro-environment, which could definitely have an impact on it and should be added to the discussion.

R: We thank the reviewer for raising this concern. Actually, in most situations, TB patients would receive anti-TB drug therapy rather than directly adopting surgical treatment, which makes it difficult for us to collect a sufficient number of untreated TB lung samples within a

reasonable length of time, thus we included some TB patients who had failed anti-TB drug therapy before tissue sampling, and tissue samples obtained from those patients could not exclude the potential effects of TB drug treatment on TB lung microenvironment. We have pointed out this limitation in the revised “Discussion” section (in the second to the last paragraph).

3. Age was not matched in the study and should be added to the discussion since it could have an impact as well.

R: We thank the reviewer for pointing out this issue. Because the peak incidence ages of pulmonary TB and sarcoidosis are relatively younger than that of lung cancers (ATS/ERS/WASOG Committee, *Am J Respir Crit Care Med*, 1999; White *et al.*, *Am J Prev Med*, 2014; Iqbal *et al.*, *Am J Public Health*, 2018), the ages of the included TB, AD and sarcoidosis patients (37.4 ± 12.0 , 60.4 ± 10.4 and 36.4 ± 13.4 , respectively) in this study were not well matched. We have pointed out this limitation in the revised “Discussion” section (in the second to the last paragraph).

4. The number of samples used for the validation of expression by qPCR, IHC were not mention for MMP8, BRCA1, PCNA and the AA metabolism related genes.

R: We thank the reviewer for pointing out this issue. We have added the missing number of samples in the legends of revised Figs. 2 and 5 and Supplementary Figs. 4 and 7.

5. They mention lung adenocarcinoma as a chronic disease, which should be changed to terminal one.

R: We thank the reviewer for raising this issue, and we have revised the manuscript carefully to avoid describing lung adenocarcinoma as a chronic lung disease.

6. There are discrepancies between supplementary Table 1 and Table 2 regarding the NC; numbers in Signs of Calcification; Signs of fibrosis; The tables should be checked.

R: We thank the reviewer for pointing this out. We have carefully revised the content of Supplementary Tables 1 and 2 to make sure that they are correct.

Overall, the manuscript is interesting and presents mostly original findings which would be of interest to the scientific community.

R: We thank the reviewer for this encouraging comment.

Reviewer #2

Remarks to the author:

This study compared and contrasted TB, sarcoidosis, and adenocarcinoma of the lung by examining the transcriptional signatures and histologic features of tissue samples from patients

with these diseases. They found that gene expression related to ECM organization (and particularly collagen matrix remodelling), immune response, cell growth and proliferation were enriched among the 3 disease groups. They further explored matrix remodelling by staining for collagen I and collagen III in tissues samples from each disease group, and found increased expression of collagen types I and III. My main concerns are that it lacks focus and it doesn't develop a story / explore a concrete hypothesis. This reads as potentially many papers in 1: 1) Distinguishing gene signatures across pulmonary diseases (this has been done previously - so would have to have a novel bent, such as their use of lung tissue rather than blood as the tissue sample source); 2) Ossification in lung disease: mechanistic and morphological features

Pulmonary remodelling in TB, sarcoid and AD; 3) TB infection and tumorigenesis.

R: We thank the reviewer for raising this issue. As described in the first two paragraphs of the manuscript, TB infection has long been linked to lung cancer (such as AD) and sarcoidosis, and the underlying mechanism remains not fully understood. Meanwhile, pulmonary TB could mimic lung cancer (Pitlik *et al.*, *Am. J. Med.*, 1984; Singh *et al.*, *Asian Pac. J. Cancer Prev.*, 2009; Falagas *et al.*, *QJM*, 2010; Shu *et al.*, *J. Clin. Med.*, 2019) or sarcoidosis (Gupta *et al.*, *Curr. Opin. Pulm. Med.*, 2012; Agrawal *et al.*, *Tuberculosis*, 2016) in some cases, which may challenge the diagnosis and delay the treatment. Therefore, in this study, we performed comprehensive analysis of both the lung molecular similarities and differences among TB, AD and sarcoidosis, with a primary aim to unravel both the potential pathogenic links among these diseases and the distinct molecular signatures of them (as is briefly described in line 59-66):

For investigation of the pathogenic links, we revealed the molecular similarities among TB, AD and sarcoidosis based on lung transcriptomes (Fig. 1). We then performed molecular experiments using additional samples to further confirm those omics-derived results (Fig. 2). Thereafter, we noticed and focused on an important pathogenic mediator (*MKI67*) shared by TB and AD to exemplify how *Mtb* infection could influence tumor cells via this molecule (Fig. 3).

In parallel, for investigation of the pathogenic differences, we revealed the distinct modular signatures of TB, AD and sarcoidosis based on lung transcriptomic data (Fig. 4). We then performed experiments using additional samples to further verify the specific molecular markers derived from modular analysis (Fig. 5). Thereafter, we focused on an important TB-specific lung signature—ossification, to investigate the underlying mechanisms (Fig. 6).

To summarize, this is an interdisciplinary and comprehensive study combining bioinformatic analysis, experimental data and clinical findings to reveal both pathogenic links and differences among three important lung diseases (Supplementary Fig. 12). We have revised the manuscript to make it more logically understandable.

The background information could be improved: The authors state that, for AD and sarcoidosis, or

for AD and TB, “discriminating these diseases could be challenging”. This is incorrect. This has not been reported, and as a respirologist, I have not confused the clinical presentations of AD and sarcoidosis, or AD and TB. The references used to support this claim also do not address diagnostic confusion for AD and these granulomatous diseases.

R: We thank the reviewer for pointing out this issue and we are sorry for making this misunderstanding. Actually, it is not that difficult to differentiate between the three diseases in most cases, but it is difficult to differentiate between these diseases in some cases, especially those share similar clinical characteristics. We have revised our manuscript accordingly. In this study, we selected patients with typical characteristics and definite diagnosis of each disease. Specifically, the included TB, AD or sarcoidosis patients were definitively diagnosed by the specialists in Beijing Chest Hospital according to the diagnostic criteria, and all sample types were verified by a certified pathologist, as described in the “Methods” section (line 440-455). We also added information regarding the likelihood of misdiagnosing of these three lung diseases (line 367-378).

Method concerns:

1. The “normal control” tissue is non-affected lung tissue from disease patients. The is not a valid control group, unless they wanted to compare, for each patient, disease and non-diseased lung. Also it wasn’t clear how many control samples were derived from each disease category. The suppl. data does not identify this.

R: We thank the reviewer for raising this issue. We included the required information in the “Methods” section (in the first paragraph of this section). As to the control group, since lung biopsy tissues from healthy people were not available, we thus used disease-uninvolved lung tissues from patients as controls. We have pointed out this limitation in the revised “Discussion” section (in the second to the last paragraph).

2. Some of the findings are an overstretch:

“In conclusion, identification of these potential pathogenic mediators of AD or SA shared by TB indicates the potential causal links between TB infection and lung AD or SA.” This is unfounded; shared pathways alone does not imply shared aetiology. In the discussion they use the term “pathogenic link” which is more appropriate.

R: We thank the reviewer for pointing this out. We have revised the manuscript accordingly to the reviewer’s suggestions (line 135-138).

“These findings indicate that Mtb promotes the proliferation, migration and invasion of A549 cells partially depending on MKI67, which mechanism might also benefit its own dissemination within the host.” This is a major statement and could be a major finding if they developed a line of investigation around it.

R: We thank the reviewer for pointing this out. We have performed more experiments

(revised Fig. 3 and Supplementary Fig. 1) to improve this part of the study (line 142-168) and we also revised our “Discussion” part accordingly (line 335-348).

3. The finding that “all these diseases drove pulmonary remodeling with various degrees of matrix destruction and fibrous protein deposition” is not surprising but has not before been adequately evaluated or definitely established. However, the interpretation that “collagen matrix remodeling in the lung is a common pathogenic characteristic of pulmonary TB, AD and SA.” might (and very likely does!) apply more broadly to pulmonary disease. Other disease groups would have been a good control here.

R: As pointed out by the reviewer, collagen matrix remodeling in the lung might apply more broadly to pulmonary disease. We thus revised our “Discussion” according to the reviewer’s suggestions as follows: “It should be noted that apart from TB, AD and sarcoidosis, some other lung diseases such as chronic obstructive pulmonary disease, idiopathic pulmonary fibrosis and asthma may also exhibit disordered lung matrix organization (Gu *et al.*, *Matrix Biol.*, 2018), and the underlying mechanistic links among those diseases warrant further investigation.” We also agree with the reviewer that other disease groups would have been a good control here. However, in this study, we focused on pulmonary TB as well as the other two potentially related lung diseases, AD and sarcoidosis, to reveal their pathogenic links and differences.

4. The figures are generally very good. Some were even stunning. But it wasn’t always clear why certain data over others was presented. Scale bars are missing for several histology figures. Small point, for reviewers it would be easier to read the figure legends if they were placed near the figure, and if the figures were labeled. The data for the “Transwell migration and invasion assays” in figure 3d is not clearly demonstrated as currently presented.

R: We thank the reviewer for raising this issue. We have revised the manuscript for better understanding. We have added the missing scale bars in Figs. 2f, 5c, h, 6b, l, and have rearranged the Fig. 3 (some results were moved into revised Supplementary Fig. 1). In addition, we have placed the figure legends behind each of the corresponding figures according to the reviewer’s suggestion.

Minor points:

1. CXR stages in sarcoidosis are not based on the 1999 statement but rather are Scadding stages.

R: We thank the reviewer for pointing this out. We have noticed that this staging system is based on a system devised by Scadding J.G. in 1961 (Scadding, *Br. Med. J.*, 1961), and we have revised our manuscript with appropriate references (line 454-455).

2. No need to abbreviate sarcoidosis (to SA) - this is not standard in the literature.

R: We thank the reviewer for this suggestion. We have revised the manuscript to avoid using

this abbreviation.

3. AD is not typically described as or considered to be a “widespread chronic disease”.

R: We have revised the manuscript according to the reviewer’s suggestion (line 45-50).

4. It wasn’t clear who read the slides of tissue sections. A pathologist expert in these diseases?

R: We thank the reviewer for raising this question. The histological and immunohistochemical analysis were performed by two trained technologists in Core Facility for Protein Research, Institute of Biophysics, Chinese Academy of Sciences, and were verified by a pathologist in Beijing Chest Hospital. We have added a brief description into “Histology and immunohistochemical analysis” in the “Methods” section (line 671-673).

5. With regards to the H37Rv MTb strains used - it would be good practice to acknowledge the source of the strains.

R: We thank the reviewer for this question. The strains of *M. tuberculosis* H37Rv used in this study are from ATCC (Catalog No. 27294). We have added this information into “Bacterial strains” in the “Methods” section (line 563).

6. The validity of using freshly thawed frozen stock to infect cells is controversial (bacteria are not going to be actively replicating and at a great level of fitness). Generally, infection occurs in exponential phase of growth.

R: We thank the reviewer for raising this concern and we are sorry for making this misunderstanding. Actually, we prepared the mycobacterial strains based on a standard experimental procedure (Kremer *et al.*, *Mycobacterium tuberculosis* protocols, 2001). Basically, bacteria were grown to mid-logarithmic phase for infection experiments, and some of the log-phase cultures were flash frozen and stored at -80°C for further repeated experiments. We have revised the “Methods” section accordingly (line 597-600). Throughout the study, to make sure that our data are credible, we conducted each of the infection experiments with appropriate controls and repeated them independently for at least three times, and we consistently observed similar results from experiments using newly prepared strains and those using freshly thawed frozen Mtb strains. Since the reviewer raised this concern, we further repeated several key infection experiments as in Fig. 6h–j using newly prepared or freshly thawed frozen strains to further determine the effect of frozen stock on the results. As shown below, the results from two parallel experiments are comparable. Thus, we believe that our conclusions derived from the infection experiments in this study are credible.

BMP/SMAD/RUNX2 pathway is required for macrophage controlling Mtb intracellular survival. a, d Survival of Mtb in U937 cells. Cells were treated with 500 nM LDN-193189, 1 μ M K02288, 10 μ M Galunisertib or control DMSO, and infected with newly prepared (a) or freshly thawed frozen (d) Mtb for 0–48 h. b, e ELISA of TNF from U937 cells treated with LDN-193189 or control DMSO and infected with newly prepared (b) or freshly thawed frozen (e) Mtb for 48 h. c, f ELISA of iNOS in U937 cells treated with LDN-193189 or control DMSO and infected with newly prepared (c) or freshly thawed frozen (f) Mtb for 48 h. $P > 0.05$, not significant (ns); * $P < 0.05$; ** $P < 0.01$; *** $P < 0.001$; **** $P < 0.0001$ (mean \pm s.e.m. of $n = 4$ in a and d, and $n = 3$ in b, c, e and f, two-way ANOVA).

Reviewer #3

Remarks to the author:

Overall this is a substantial and well-presented article that provides important insights into the molecular mechanisms underlying the link between lung calcification and ossification in TB, as well as the identification of potential markers to better distinguish between TB, AD and SA.

R: We thank the reviewer for the encouraging comments on our manuscript.

However, I do not find the initial data (Figs 1 and 2) particularly novel. It is not surprising to me that three diseases well known to cause significant damage to the lung would have similar molecular signatures related to immune response, metabolic processes and pulmonary remodeling, or that signature genes between TB and AD involved in cell growth and proliferation were most directly correlated with tumor development. Similarly, the involvement of MKI67 is not surprising in this context.

R: We thank the reviewer for pointing out this issue. Although each disease of pulmonary TB, AD and sarcoidosis could cause tissue damage, and thus was speculated to be similarly associated with immune response, metabolic processes and pulmonary remodeling, evidence derived from unbiased comparison and verification of human lung-based molecular signatures of these diseases is still lacking. Also, despite of those speculation, it was not quite clear that what pathogenic molecules and pathways in the lung are potentially shared by these diseases, and which of them could be most predominant. Therefore, we performed Figs. 1, 2 to compare and verify the lung molecular similarities among these three important lung diseases, which have not been comprehensively clarified.

Likewise, the shared genes and pathways between TB and AD in this study were screened out unbiasedly. Actually, there were 339 overlapped signature genes shared by TB and AD obtained from genome-wide lung transcriptional profiling (Supplementary Data 2, 3), from

which we further screened out the most important 65 TB-AD-shared genes using TCGA data (Fig 1c, d). Notably, these 65 genes were screened out based on Kaplan-Meier survival analysis, and thus is most significantly correlated with the overall survival of lung AD patients (Supplementary Data 5). Therefore, they were probably the key AD-correlated pathogenic mediators that might be affected by TB infection, and thus could provide important clues for the potential molecular mechanisms underlying the pathogenic link among pulmonary TB and AD (as described in line 96-98 and line 103-105).

Additionally, results from Figs. 1, 2 are necessary for making this study logically understandable. As described previously (the answer to the Remarks from the reviewer #2), this is a comprehensive study to unbiasedly investigate both pathogenic links (Figs. 1–3) and differences (Figs. 4–6) among TB, AD and sarcoidosis (as briefly introduced in line 59-66). To exploit pathogenic links among diseases, we first revealed the molecular similarities among TB, AD and sarcoidosis based on lung transcriptomes (Fig. 1), and then performed molecular experiments using additional samples to further verify those omics-derived results (Fig. 2). Through these two steps, we confirmed the credibility of the lung transcriptome-derived data, and from these results we can screen out and focus on important pathogenic mediators (such as *MKI67*, Fig. 3) to investigate how *Mtb* manipulates those potential pathogenic mediators to facilitate the corresponding disease outcomes. Taken together, we suggest that Figs. 1, 2 are not unnecessary.

However, the idea of *Mtb* directly promoting the proliferation, migration and invasion of human lung adenoma A549 cells is interesting (Fig 3), but the authors use a CCK-8 assay to measure cell proliferation and this only determines cell viability, a proper cell proliferation assay should be performed. This part of the study could also be improved, especially considering the rest of the article is better developed. To add strength to this section it would be beneficial if the authors could show that *Mtb*-secreted PtpA regulates *MKI67* in the A549 cells.

R: We thank the reviewer for this valuable suggestion. We further confirmed our data from CCK-8 assay through conducting cell proliferation assay using 5-ethynyl-2'-deoxyuridine (EdU), a nucleoside analogue which can be incorporated into the DNA of proliferating cells (Fig. 3a, b). As suggested by the reviewer, we further examined the role of PtpA in *MKI67*-mediated cell proliferation, migration and invasion during *Mtb* infection. And we found that deletion of PtpA in *Mtb* largely attenuated the effect of *Mtb* infection-induced upregulation of *MKI67* mRNA (Fig. 3g) and Ki-67 protein in A549 cells (Fig. 3h, i). Consistently, PtpA deletion also markedly abolished the effects of *Mtb* infection-promoted A549 cell proliferation, migration and invasion, which were largely dependent on Ki-67 (Supplementary Fig. 1d–j). Together, these results indicate that *Mtb* effector protein PtpA contributes to *Mtb*-promoted tumor cell proliferation, migration and invasion, which are partially dependent on *MKI67*. We also revised the “Discussion” accordingly (line 335-348).

The discussion should include a section “limitations of the study”: potential points to discuss: 1) the caveat/strength of normal controls; they are actually derived from the patients. 2) the authors argue consistently that it is very difficult to differentiate between the three diseases, but they by themselves obviously did not have problems doing so; how high is the likelihood of misdiagnosing?

R: We thank the reviewer for this constructive suggestion. We have added the limitations of the study in the “Discussion” section (line 421-428).

As to the difficulties of differentiation diagnosis between the three diseases, we are sorry for making the misunderstanding. Actually, it is not that difficult to differentiate between the three diseases in most cases, but it is difficult to differentiate between these diseases in some cases, especially those share similar clinical characteristics. We have revised our manuscript accordingly. In this study, we selected patients with typical characteristics and definite diagnosis of each disease. Specifically, the included TB, AD or sarcoidosis patients were definitively diagnosed by the specialists in Beijing Chest Hospital according to the diagnostic criteria, and all sample types were verified by a certified pathologist, as described in the “Methods” section (line 440-455). We also added information regarding the likelihood of misdiagnosing of these three lung diseases (line 367-378).

Once again, we greatly appreciate the reviewers for having helped us improve this manuscript tremendously.

REVIEWERS' COMMENTS:

Reviewer #2 (Remarks to the Author):

The authors did a nice job of addressing the comments from the first review. The manuscript reads better, and is more complete. I applaud their efforts.

I remain concern about the mis-leading statement "Pulmonary tuberculosis (TB), which is caused by Mycobacterium tuberculosis (Mtb), has long been linked to the development of lung adenocarcinoma (AD) and sarcoidosis". TB is associated with an increased risk of lung cancer, although it's not clear that adenocarcinoma is the most common type. At the very least, the manuscript should not imply (in the first line of the abstract, and also lines 42-43) that TB causes sarcoidosis - that is certainly not a widely held belief and is not defended by currently available data. Identifying mycobacterial (which is not MTB specific) genetic material in sarcoid lesions is not the same thing as finding evidence that MTB causes sarcoidosis.

Minor suggestion, but I would avoid the word "probably" (line 55, 98, 116, 120, 211). It's too assuming. The word "perhaps" might work better for some of these.

There are a few minor edits that will be picked up by the journal team, but overall the writing is excellent - both in style and in "story telling" of why each step was done, and how each finding fits into the larger picture of the diseases.

Point-by-point responses to reviewers' comments

Chai, et al., “**Lung gene expression signatures suggest pathogenic links and molecular markers for pulmonary tuberculosis, adenocarcinoma and sarcoidosis**” (COMMSBIO-20-0930A).

REVIEWERS' COMMENTS:

Reviewer #2 (Remarks to the Author):

The authors did a nice job of addressing the comments from the first review. The manuscript reads better, and is more complete. I applaud their efforts.

I remain concern about the mis-leading statement "Pulmonary tuberculosis (TB), which is caused by *Mycobacterium tuberculosis* (Mtb), has long been linked to the development of lung adenocarcinoma (AD) and sarcoidosis". TB is associated with an increased risk of lung cancer, although it's not clear that adenocarcinoma is the most common type. At the very least, the manuscript should not imply (in the first line of the abstract, and also lines 42-43) that TB causes sarcoidosis - that is certainly not a widely held belief and is not defended by currently available data. Identifying mycobacterial (which is not MTB specific) genetic material in sarcoid lesions is not the same thing as finding evidence that MTB causes sarcoidosis.

Minor suggestion, but I would avoid the word "probably" (line 55, 98, 116, 120, 211). It's too assuming. The word "perhaps" might work better for some of these.

There are a few minor edits that will be picked up by the journal team, but overall the writing is excellent - both in style and in "story telling" of why each step was done, and how each finding fits into the larger picture of the diseases.

R: We thank the reviewer for pointing out these issues, and we have modified the abstract (according to the edits suggested by the editors) and the main text accordingly.

Once again, we greatly appreciate the reviewer for having helped us improve this manuscript tremendously.